# Horizontally acquired *papGII*-containing pathogenicity islands underlie the emergence of invasive uropathogenic *Escherichia coli* lineages

Michael Biggel [1✉], Basil B. Xavier [1], James R. Johnson[2], Karen L. Nielsen [3], Niels Frimodt-Møller[3], Veerle Matheeussen [1,4], Herman Goossens[1,4], Pieter Moons[1,6] & Sandra Van Puyvelde [1,5,6✉]

*Escherichia coli* is the leading cause of urinary tract infection, one of the most common bacterial infections in humans. Despite this, a genomic perspective is lacking regarding the phylogenetic distribution of isolates associated with different clinical syndromes. Here, we present a large-scale phylogenomic analysis of a spatiotemporally and clinically diverse set of 907 *E. coli* isolates, including 722 uropathogenic *E. coli* (UPEC) isolates. A genome-wide association approach identifies the (P-fimbriae-encoding) *papGII* locus as the key feature distinguishing invasive UPEC, defined as isolates associated with severe UTI, i.e., kidney infection (pyelonephritis) or urinary-source bacteremia, from non-invasive UPEC, defined as isolates associated with asymptomatic bacteriuria or bladder infection (cystitis). Within the *E. coli* population, distinct invasive UPEC lineages emerged through repeated horizontal acquisition of diverse *papGII*-containing pathogenicity islands. Our findings elucidate the molecular determinants of severe UTI and have implications for the early detection of this pathogen.

[1] Laboratory of Medical Microbiology, Vaccine and Infectious Disease Institute, University of Antwerp, Antwerp, Belgium. [2] Veterans Affairs Medical Center and University of Minnesota, Minneapolis, MN, USA. [3] Department of Clinical Microbiology, Rigshospitalet, Copenhagen, Denmark. [4] Laboratory of Clinical Microbiology, Antwerp University Hospital, Antwerp, Belgium. [5] Cambridge Institute of Therapeutic Immunology and Infectious Disease (CITIID), Department of Medicine, University of Cambridge, Cambridge CB2 0SP, UK. [6] These authors contributed equally: Pieter Moons, Sandra Van Puyvelde. ✉email: michaelbiggel@gmail.com; Sandra.VanPuyvelde@uantwerpen.be

In humans, *Escherichia coli* is both a commensal and a pathogen capable of causing intestinal and extraintestinal disease. In the urinary tract, *E. coli* causes a wide range of clinical syndromes that includes, from less to more severe, asymptomatic bacteriuria (ABU), bladder infection (cystitis), kidney infection (pyelonephritis), and urinary-source bacteremia. An estimated half of all *E. coli* bloodstream infections originate from the urinary tract[1–3].

*E. coli* strains with a special ability to cause urinary-tract infection (UTI) are classified as uropathogenic *E. coli* (UPEC). UPEC has further been grouped with *E. coli* strains that cause (non-urinary-source) bacteremia, meningitis, and prostatitis as extraintestinal pathogenic *E. coli* (ExPEC)[4]. UPEC are considered to be opportunistic pathogens, with the human gut as their reservoir[5]. The different pathotypes of intestinal pathogenic *E. coli* (enteropathogenic *E. coli* [EPEC], enterohemorrhagic *E. coli* [EHEC], enterotoxigenic *E. coli* [ETEC], enteroinvasive *E. coli* [EIEC], and enteroaggregative *E. coli* [EAEC]) are differentiated genotypically by their characteristic virulence genes. In contrast, *E. coli* isolates are often classified presumptively as ExPEC and/or UPEC according to their site of infection and isolation, irrespective of their intrinsic virulence, which usually is unknown[6,7].

Although multiple urovirulence factors and their importance in UTI pathogenesis have been described, genotypical features that distinguish UPEC from non-pathogenic *E. coli* remain incompletely defined[5]. The identification of defining UPEC features is hampered by the large number of potentially redundant bacterial virulence-associated genes (VAGs), a shared habitat with commensal *E. coli* in the intestinal microbiota, and varying host susceptibility to UTI[8]. UTI pathogenesis is hence assumed to be determined by a complex interplay of bacterial and host factors.

The *E. coli* population is subdivided into phylogenetic groups (phylogroups) A, B1, B2, C, D, F, and G, and cryptic clades, with ExPEC and UPEC strains deriving predominantly from phylogroups B2 and D. Strains from phylogroups B2 and D typically have more VAGs than do those from phylogroups A and B1, which are associated with commensal and intestinal pathogenic *E. coli*[4,9,10]. Phylogroups B2 and D encompass the pandemic UPEC lineages clonal complex (CC) 69, CC73, CC95, and CC131, which are responsible for most cases of *E. coli* cystitis, pyelonephritis, and bloodstream infection worldwide. This predominance in extraintestinal *E. coli* infections of just a few *E. coli* lineages, among the hundreds that exist, suggests that specific genetic determinants facilitate the expansion, global spread, and virulence of these pathogens[11]. Increased antimicrobial resistance (AMR) might have contributed to the recent dissemination of CC69 and CC131, whereas it cannot explain the epidemiologic success of the largely susceptible lineages CC73 and CC95[12–14].

Previous phylogenomic analyses provided important insights into the population structure, virulence associations, and AMR of ExPEC and UPEC[12,13,15–18]. However, a specific genomic analysis of invasive UPEC strains—defined here as those associated with severe UTI (pyelonephritis or urinary-source bacteremia)—is lacking. PCR-based studies targeting subsets of VAGs have identified a greater frequency of various VAGs in pyelonephritis and urinary-source bacteremia isolates, as compared to ABU, cystitis, or fecal isolates[19–22]. These VAGs include *papGII*, which encodes one of several PapG tip adhesin variants of P fimbriae. PapGII binds to the globoseries of glycosphingolipids on uroepithelial cells and transcriptionally regulates host gene expression in kidney cells, leading to a pyelonephritis-associated IRF-7 response[23]. Like many other *E. coli* VAGs, the *pap* operon encoding P fimbriae lies on pathogenicity islands (PAIs)[24]. PAIs are large horizontally transferable genetic elements assumed to play an important role in the evolution of pathogenic *E. coli*[25].

In this study, we used a genomics approach to investigate the population structure, virulence determinants, and evolution of invasive UPEC isolates, as compared to non-invasive UPEC isolates, which were defined here as those associated with asymptomatic bacteriuria (ABU) or cystitis. Fecal isolates not associated with disease were included to investigate their genetic similarity with UPEC. Our analysis identified a small number of enduring and broadly distributed invasive UPEC lineages that seemingly emerged after independent horizontal acquisitions of *papGII*.

## Results

**Invasive UPEC cluster in distinct phylogenetic lineages**. We analyzed whole-genome sequences of 722 total UPEC isolates— including 385 invasive UPEC isolates (from five collections) and 337 non-invasive UPEC isolates (from nine collections)—and 185 fecal isolates (from two collections). The 722 UPEC isolates were so defined based on their source of isolation and associated clinical presentation, whereas the fecal isolates were not associated with UTI or intestinal infection (Supplementary Note 1 gives the various source collections' inclusion criteria). The 16 isolate collections had a broad spatiotemporal distribution: they derived from various locales in Europe and the United States and spanned nearly four decades (1981–2018) (Supplementary Fig. 1, Table 1). For four of these collections (LtABU, MC_pye, MVAST_ABU, UZA_uro), the isolates ($n = 151$) were sequenced within this study; for the remaining 12 collections ($n = 756$ isolates) the sequence data were publicly available (Supplementary Table 1).

Phylogenetic analysis revealed a non-random phylogenetic distribution of the three clinical phenotypes, i.e., invasive UPEC, non-invasive UPEC, and fecal isolates. Invasive and non-invasive UPEC isolates alike were predominantly (86% and 71%, respectively) from phylogroups B2 and D; by contrast, fecal isolates were more evenly distributed among phylogroups A, B1, B2, and D (Supplementary Table 2, Supplementary Note 2). Among the invasive UPEC isolates the 11 dominant CCs were CC12, CC14, CC23, CC31, CC62, CC69, CC73, CC95, CC131, CC144, and CC405 (the corresponding sequence types (STs) are shown in Fig. 1). These 11 dominant CCs accounted collectively for 82% of all invasive UPEC isolates, with limited variation between collections (Supplementary Fig. 2). Moreover, within these CCs, invasive and non-invasive UPEC isolates tended to cluster into distinct sublineages (Fig. 1). Most of these CCs encompassed one or multiple sublineages that were significantly enriched with invasive UPEC isolates ($P < 0.05$, for proportion of invasive UPEC isolates within vs. outside the sublineage; Supplementary Data 2), giving 12 sublineages associated with invasiveness. Within these 12 sublineages combined, 86% of all UPEC isolates were invasiveness-associated, vs. 39% of other UPEC isolates.

**Invasive UPEC lineages emerge after acquisition of *papGII*+ PAIs**. A pan-genome-wide association study (pan-GWAS) comparing invasive vs. non-invasive UPEC isolates that included 30,705 clusters of orthologous genes (COGs) identified significant associations with invasiveness for two genomic loci, *papGII* and *iuc*, which encodes aerobactin biosynthesis (Fig. 2, Table 2). An alternative, gene-prediction-independent GWAS approach based on De Bruijn graphs and linear mixed models (DBGWAS[26]) confirmed *papGII* as the most significantly invasiveness-associated genomic region (Supplementary Table 3, Supplementary Data 3). Overall, *papGII* was present in 63.9% of invasive UPEC isolates, as compared with 15.4% of non-invasive UPEC isolates (OR = 9.7, $P = 6.0 \times 10^{-42}$) and, for reference, 19% of fecal isolates (OR = 7.6, $P = 9.1 \times 10^{-25}$).

**Table 1 Isolate collections included in this study.**

| Collection | Clinical source | No. isolates | Host age (median) | Country of isolation | Year of isolation | Reference |
|---|---|---|---|---|---|---|
| LtABU | Long-term ABU (non-invasive UPEC) | 43 | 70–97 (86) | Belgium | 2017–2018 | This study |
| dsABU | ABU (non-invasive UPEC) | 9 | 12–25 (19) | Hungary | 2010–2012 | Stork et al.[112] |
| RT_ABU | ABU (non-invasive UPEC) | 19 | 29–74 (54) | Belgium | 2012–2015 | Coussement et al.[113] |
| MVAST_ABU | ABU (non-invasive UPEC) | 39 | 38–92 (69) | USA | 2010–2011 | This study; Drekonja et al.[69] |
| Koege_cys | Cystitis (non-invasive UPEC) | 19 | 3–81 (61) | Denmark | 2005–2006 | Skjøt-Rasmussen et al.[114] |
| KTE_cys | Cystitis (non-invasive UPEC) | 48 | 19–53 (34) | Denmark | 2009–2010 | Nielsen et al.[18] |
| PUTI_cys | Cystitis (non-invasive UPEC) | 30 | na | USA | 1999–2000 | Johnson et al.[72]; Sannes et al.[70] |
| UMEA_cys | Cystitis (non-invasive UPEC) | 105 | 17–85 (48) | Sweden | 1995–1997 | Ejrnæs et al.[115] |
| Rec_cys | Recurrent cystitis (non-invasive UPEC) | 15 | 18–49 (na) | USA | 2003–2006 | Czaja et al.[116], Schreiber et al.[43] |
| MC_pye | Acute uncomplicated pyelonephritis (invasive UPEC) | 70 | >18 (na) | USA | 1994–1997 | This study; Johnson et al.[72]; Sannes et al.[70,117]; Talan et al.[71] |
| HVH_urb | Urinary-source bacteremia (invasive UPEC) | 190 | 19–102 (79) | Denmark | 2003–2005 | Skjøt-Rasmussen et al.[118] |
| UHS_urb | Urinary-source bacteremia (invasive UPEC) | 22 | 19–96 (70) | UK | 2015–2016 | Dale et al.[119] |
| BUTI_uro | Urinary-source bacteremia (invasive UPEC) | 67 | 20–91 (62) | USA | 1981–1985 | Johnson et al.[40,72] |
| UZA_uro | Urosepsis (invasive UPEC) | 30 | 0–92 (75) | Belgium | 2015–2017 | This study |
| KTE_fec | Fecal (non-disease associated) | 102 | 18–53 (37) | Denmark | 2009–2010 | Nielsen et al.[18] |
| MN_fec | Fecal (non-disease associated) | 81 | na | USA | 1996–2000 | Johnson et al.[72]; Sannes et al.[70,120] |

*ABU asymptomatic bacteriuria, UPEC uropathogenic E. coli.*

This significant association of *papGII* with invasive UPEC was found irrespective of host gender or age (Supplementary Table 4). In addition, within each clinical phenotype (invasive UPEC, non-invasive UPEC, and fecal) the frequency of *papGII* varied only modestly between collections, ranging by collection from 58 to 77% for invasive UPEC, vs. from 0 to 29% for non-invasive UPEC, and 19 to 20% for fecal isolates; the observed modest variation likely reflected the collections' differing settings and patient inclusion criteria (Supplementary Fig. 3). The presence of *papGII* ($n = 333$ isolates) coincided closely with the above-defined invasive UPEC-enriched sublineages (Fig. 1); 88% of isolates in these sublineages carried *papGII*, vs. 19% of other isolates ($P < 0.001$).

The 333 *papGII*-positive (*papGII*+) isolates clustered robustly in 14 discrete lineages, which thus were termed *papGII*+ lineages (Fig. 3a). CC12, CC14, CC31, CC59, CC62, CC69, CC95, CC131, CC144, and CC405 encompassed one such *papGII*+ sublineage each. CC73 showed a heterogenous population structure that encompassed four distinct *papGII*+ lineages (called L1–L4), as identified by using a second hierarchic level of clustering. Most *papGII*+ lineages comprised isolates from different regions that had been collected during different time intervals over 40 years, suggesting enduring, broadly distributed lineages rather than transient and/or geographically restricted subpopulations (Supplementary Fig. 2).

The genetic context of *papGII* in the 14 *papGII*+ lineages was usually unclear from short-read sequencing data only, but could be resolved for 35 isolates, including at least one from each *papGII*+ lineage. These 35 genomes comprised 14 publicly available, high-quality assemblies and 21 complete or near-complete assemblies obtained *in-house* using long-read sequencing (Supplementary Data 5). Six of the 35 genomes harbored more than one *papGII* locus. The total of 42 *papGII* genes were each part of a complete *pap* operon (*papGII* operons) that consisted of 11 genes (*papIBAHCDJKEFG*), with sequence variation found for *papA* (major fimbrial subunit) and *papE* (minor fimbrial subunit) (Supplementary Fig. 4) as described previously[27,28]. The *papGII* genes always occurred on PAIs, i.e., genomic regions flanked by an integrase, integrated into the chromosome at one of six specific loci, and absent in phylogenetically related isolates.

These 42 resolved *papGII*-containing (i.e., *papGII*+) PAIs were integrated directly downstream of the tRNA genes *pheV*, *pheU*, *selC*, or *leuX*, or into the *gln* or *ula* operons (Supplementary Fig. 5). In addition, the *papGII*+ PAI insertion site could be determined in 170 other *papGII*+ isolates (171 putative PAIs) for which only short-read assemblies were available. Overall, the tRNA-*pheV* and tRNA-*pheU* sites accounted for 194 (91%) of all 213 identified *papGII*+ PAI insertion sites; the remaining 9% were divided between four other tRNA sites, which accounted for 0.5–4% of the PAIs each (Supplementary Data 7). PAIs inserted at the (predominant) tRNA-*pheV* and tRNA-*pheU* sites shared the same integrase gene, whereas those inserted at the other four sites had distinct insertion-site-specific integrase sequences (Supplementary Fig. 6). Complete or near-complete assemblies of three *papGII*+ isolates each from CC144 and CC12 revealed the sporadic presence of two or three *papGII* operons, each on a distinct PAI (Supplementary Fig. 7).

The 42 resolved *papGII*+ PAIs showed a highly diverse gene content: the only consistently present element was the *papGII* operon (Fig. 3b, Supplementary Fig. 8a). This is in line with the concept that PAIs are organized in mosaic-like structures, consisting of a flexible pool of gene modules[29,30]. The *papGII*+ PAIs ranged in size from 28 kb (26 genes) to 146 kb (140 genes). They contained from 12 to 31 VAGs each (including the 11 *pap* genes), which encoded up to seven virulence factors per PAI

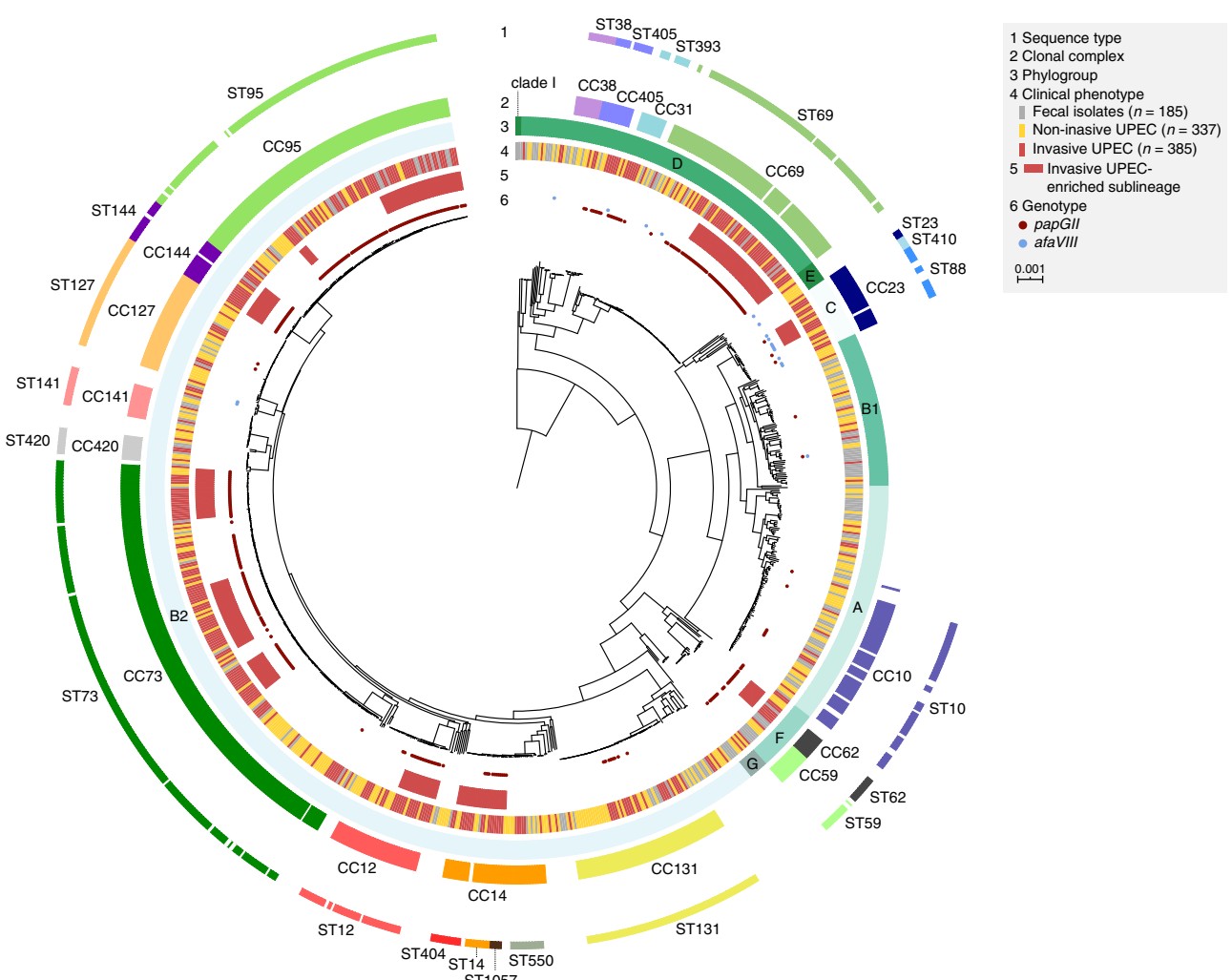

**Fig. 1 Phylogeny of 907 *Escherichia coli* isolates associated with different clinical phenotypes.** Midpoint-rooted maximum-likelihood phylogenetic tree based on 109,023 variable sites identified in a core genome alignment (1.136 Mbp). Ring 1, 2, and 3 denote predominant sequence types (ST), corresponding clonal complexes (CC), and phylogroup assignment. Clinical phenotypes are labeled according to the key (ring 4). Phylogenetic clusters identified using BAPS (Bayesian analysis of population structure) significantly enriched with invasive UPEC isolates are highlighted in ring 5. The presence of the *afaVIII* and *papGII* (blue and red dots) across the phylogeny is annotated in ring 6. The scale bar indicates the number of substitutions per site in the core genome alignment. A tree with bootstrap support values is provided in Supplementary Fig. 20. The tree was visualized using iTOL[89]. An interactive visualization of this phylogenetic tree can be found out at https://microreact.org/project/O4QAYAJWw.

(Supplementary Fig. 8a, Supplementary Table 5). They also contained multiple hypothetical genes, phage gene remnants, and IS elements, but no AMR genes. Based on sequence similarity, the 42 resolved *papGII*+ PAIs could be grouped into six types, I through VI (Supplementary Fig. 8b). The incompletely resolved *papGII*+ PAIs of 199 additional isolates could be assigned to these six types by mapping reads to the resolved PAIs (Supplementary Data 11). Within a given *papGII*+ lineage, most *papGII*+ PAIs were fully conserved or differed only by a few IS elements, suggesting a single PAI acquisition event per *papGII*+ lineage (Fig. 3a). By contrast, the sporadic occurrence of similar PAIs in distantly related isolates suggested horizontal gene transfer events (Supplementary Fig. 9).

Exceptionally, one invasive UPEC-enriched sublineage was not characterized by the presence of *papGII*: although phylogroup C (represented almost entirely by CC23 isolates: 24/25, 96%) comprised mainly invasive UPEC isolates (17/25, 68%), only three CC23 isolates contained *papGII*. Screening of CC23 isolates for known *E. coli* VAGs identified the *afaVIII* operon (*afaABCDE-VIII*) in eight (of 14) *papGII*-negative invasive UPEC

isolates (Fig. 1). *afa* genes encode afimbrial structures that mediate adhesion and invasion[31]; some of these, including AfaEVIII, have been associated with pyelonephritis[32,33]. Over the entire dataset, *afaVIII* occurred in 15 invasive UPEC isolates (3.9%), as compared with only two non-invasive UPEC isolates (0.6%; OR = 6.8, $P = 0.003$) and one fecal isolate (0.5%: OR = 7.4, $P = 0.03$). These *afaVIII* operons were usually complete and the *afa* genes were conserved, apart from minor sequence variation within *afaEVIII* (Supplementary Fig. 10). Long-read sequencing of CC23 isolate US26 resolved *afaVIII* on an 82-kb genomic island that was integrated at the tRNA-*argW* site. This hybrid pathogenicity-resistance island carried one additional VAG, *agn43*, which encodes antigen 43 (Ag43), a protein involved in biofilm formation, adhesion, and autoaggregation[34], plus 10 genes predicted to confer resistance to beta-lactams, sulfonamides, phenicols, aminoglycosides, or mercury (Supplementary Fig. 11).

**UPEC isolates harbor a diverse VAG repertoire.** The UPEC isolates each contained between 164 and 382 VAGs, some of

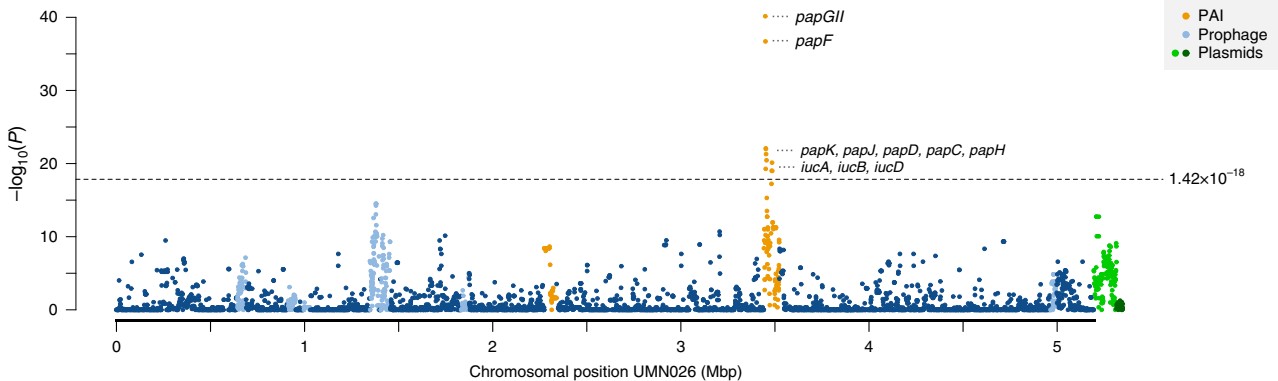

**Fig. 2 Manhattan plot for pan-genome-wide associations for invasive vs. non-invasive UPEC.** Data are based on 30,705 clusters of orthologous genes (COGs) identified in 722 UPEC isolates. Fecal isolates were not considered in this analysis because they present no urinary phenotype. The plot shows genes assigned to 4764 unique COGs identified in the genome of reference strain UMN026 (CC69). Each dot represents one COG. The vertical axis denotes raw $P$ values of Fisher's exact statistics. To account for the effects of sample size and population structure, the genome-wide significance threshold (dotted line, $P = 1.42 \times 10^{-18}$) was inferred from a simulated dataset using treeWAS. The horizontal axis gives the nucleotide position in the chromosome. COGs part of pathogenicity islands (PAIs), prophage regions, or the two plasmids of UMN026 are color-labeled, including the High Pathogenicity Island (HPI) and PAI$_{UMN026-pheV}$, a $papGII$-containing PAI. The remaining 25,941 COGs of the pan-genome that did not map to UMN026 were not pan-genome-wide significant (Supplementary Fig. 21).

which were located on $papGII+$ PAIs (Fig. 4a, Supplementary Fig. 8a). VAGs identified on one or more of the 42 resolved $papGII+$ PAIs included iron uptake systems ($iuc$, $ireA$, $iha$, $fbp$, $fec$), fimbrial genes ($fim$, $ucl$), invasins ($tia$, $hek$, $hra2$), toxins ($cnf1$, $hly$), autotransporters ($sat$, $agn43$), toxin-antitoxin modules ($cdiA/cdiB$), and $sisA$ and $sisB$, which in mice downregulate the kidney's immune response during *E. coli* infection[35].

The number of VAGs correlated with the isolates' associated disease severity. Genomes from invasive UPEC isolates contained significantly more VAGs than did those from cystitis isolates, and genomes from cystitis isolates contained significantly more VAGs than did those from ABU isolates (Fig. 4a). To account for this observation, cystitis isolates were analyzed separately from ABU isolates. To examine functional correlations, VAGs were grouped by presumptive functional class (Supplementary Data 8). With VAGs stratified by functional class, as compared with either the cystitis or ABU isolates the invasive UPEC isolates had a higher prevalence of VAGs related to iron uptake and immune evasion/ modulation. In addition, as compared with ABU isolates, they had a higher prevalence of VAGs related to secretion systems/ autotransporters, adhesion/invasion (attributable to $pap$ genes only), toxins, and bacteriocins (Supplementary Table 6a).

The number of VAGs varied significantly by phylogroup, analogous to the observed phylogroup-specific segregation of clinical phenotypes. Specifically, the number of VAGs was significantly higher in UPEC-associated phylogroups (B2, C, D, F) than in those associated with fecal isolates (A and B1) (Supplementary Fig. 12, Supplementary Table 7). Within phylogroup B2, the total number of VAGs did not differ significantly between invasive UPEC isolates and cystitis isolates, but was significantly higher among invasive UPEC isolates as compared to ABU isolates, a difference that is possibly attributable to the acquisition or loss of single PAIs (Supplementary Fig. 13, Supplementary Table 6b). By contrast, among $papGII+$ isolates the number of VAGs did not differ significantly by clinical source, i.e., between invasive UPEC, cystitis, ABU, or fecal isolates (Supplementary Table 8).

Iron acquisition systems are regarded as critical virulence factors of UPEC[36]. Of the 22 presumably partially redundant iron uptake systems described for *E. coli*[37–39], the average number of such systems per isolate was significantly higher among invasive UPEC isolates (15.6) than among cystitis isolates (14.3), ABU isolates

(14.0), or fecal isolates (12.9) (Supplementary Table 9). $papGII+$ isolates carried, on average, 16.3 iron uptake systems (vs. 13.5 among $papGII$-negative isolates, $P < 0.001$). As noted generally for VAGs, the average number of iron uptake systems per isolate correlated with phylogenetic background, with the highest values for (invasiveness-associated) phylogroups B2 and F (Fig. 4b, Supplementary Table 9). Apart from iron acquisition systems, previous studies found associations with uropathogenicity for multiple other VAGs[40–43]. In agreement with these findings, many of these uropathogenicity-associated VAGs were here found in both invasive and non-invasive UPEC; and were associated with specific CCs rather than invasiveness (Supplementary Fig. 14).

Apart from $papGII$, only the $iuc$ (i.e., aerobactin) locus, which in *E. coli* consists of six genes ($shiF$, $iucA$, $iucB$, $iucC$, $iucD$, $iutA$), reached pan-genome-wide statistical significance for invasive vs. non-invasive UPEC isolates ($P = 7.6 \times 10^{-21}$). The $iuc$ locus occurred in 73% of invasive UPEC isolates, vs. 36% of cystitis isolates ($P < 0.001$, OR = 4.7) and 48% of ABU isolates ($P < 0.001$, OR = 2.9). It also was associated specifically with $papGII$, occurring in 86% of $papGII+$ isolates vs. 33% of $papGII$-negative isolates ($P < 0.001$, OR = 13.0). A complete $iuc$ locus (100% sequence coverage) was identified in 92% of the 482 $iuc+$ isolates. In the remaining 38 $iuc+$ isolates the $iuc$ locus either could not be resolved (15 isolates) or showed disruptions (23 isolates). We identified three distinct, highly conserved $iuc$ locus architectures. These included (i) $shiF/iuc/iutA1$, which was usually associated with type II $papGII+$ PAIs; (ii) $shiFp/iuc/iutA2$, which was part of large IncFII plasmids and corresponds to the $iuc5$ locus described in *Klebsiella* isolates[44]; and (iii) $shiF/iuc/iutA2$, which was part of the widely distributed PAI$_{ABU83972-pheV}$-like islands or, less commonly, of PAI$_{IAI39-pheV}$-like islands (Supplementary Fig. 15). The distribution of the non-aerobactin iron uptake systems by clinical phenotype and phylogroup is described in Supplementary Note 3 and Supplementary Table 10.

**$papGII+$ sublineages within CC69, CC95, and CC73.** The three pandemic UPEC lineages CC69, CC95, and CC73 comprised, collectively, 42% (301/722) of the UPEC isolates and 68% (229/ 333) of the $papGII+$ isolates, as compared with 23% of the fecal isolates. Each of these lineages exhibited gain, loss, and/or rearrangements of $papGII+$ PAIs, which segregated by lineage.

**Table 2 Clusters of orthologous genes (COGs) with pan-genome-wide significant associations for invasive vs. non-invasive uropathogenic *E. coli* (UPEC).**

| Gene | Gene product | Associated locus | Frequency invasive UPEC isolates (n = 385) | Frequency non-invasive UPEC isolates (n = 337) | P | Adjusted P[a] | Odds ratio (95% CI) |
|---|---|---|---|---|---|---|---|
| papGII/[b] | P fimbriae adhesin variant PapGII | papGII | 246 (63.9%) | 52 (15.4%) | 6.0E-42 | 1.7E-37 | 9.7 (6.7-14.2) |
| papF | P fimbriae minor subunit PapF | papGII | 230 (59.7%) | 49 (14.5%) | 2.0E-37 | 5.6E-33 | 8.7 (6.0-12.8) |
| papJ | P fimbriae assembly protein PapJ | various pap | 285 (74.0%) | 128 (38.0%) | 8.4E-23 | 2.4E-18 | 4.6 (3.3-6.5) |
| papD | P fimbriae chaperone PapD | various pap | 290 (75.3%) | 133 (39.5%) | 1.0E-22 | 2.9E-18 | 4.7 (3.4-6.5) |
| papC | P fimbriae outer membrane usher PapC | various pap | 288 (74.8%) | 133 (39.5%) | 5.1E-22 | 1.4E-17 | 4.5 (3.3-6.3) |
| papH | P fimbriae minor subunit PapH | various pap | 284 (73.8%) | 132 (39.2%) | 3.6E-21 | 1.0E-16 | 4.4 (3.1-6.1) |
| iucB | Aerobactin biosynthesis protein IucB | iuc | 283 (73.5%) | 132 (39.2%) | 7.6E-21 | 2.2E-16 | 4.3 (3.1-6.0) |
| papK | P fimbriae minor subunit PapK | various pap | 274 (71.2%) | 126 (37.4%) | 5.3E-20 | 1.5E-15 | 4.1 (3.0-5.7) |
| iucC | Aerobactin biosynthesis protein IucC | iuc | 284 (73.8%) | 136 (40.4%) | 9.4E-20 | 2.7E-15 | 4.1 (3.0-5.8) |
| iucA | Aerobactin biosynthesis protein IucA | iuc | 283 (73.5%) | 135 (40.1%) | 1.0E-19 | 2.9E-15 | 4.1 (3.0-5.8) |

Frequencies, P values (Fisher's exact test), and odds ratios with 95% confidence intervals (CI) are shown for COGs with P values below the simulation inferred significance threshold (raw $P = 1.42 \times 10^{-18}$ or Bonferroni adjusted $P = 4.04 \times 10^{-14}$). Frequencies of papGII, which was occasionally fragmented in assemblies due to the presence of multiple papG alleles, were corrected using read-mapping-based identification (Supplementary Data 9). Other pap genes (papIBAHCDJKEF) were identified at lower significance levels than papGII, because these co-occurred in combination with other, non-significant papG alleles (papGI, papGIII, papGIV, papGV) or occurred as different variants. The iuc locus in E. coli consists of six genes: shiF, iucABCD, iutA. The genes shiF and iutA occurred in two alleles (shiF, shiFp; iutA1, iutA2; Supplementary Fig. 15) assigned to distinct COGs, resulting in decreased pan-genome-wide significance. Additional COGs with P values below the Bonferroni adjusted P = 0.05 (corresponding to a raw $P = 1.76 \times 10^{-6}$) are provided in Supplementary Data 4.
[a]Bonferroni adjusted for comparisons of 28,468 candidate COGs.
[b]Frequencies corrected by read mapping; uncorrected BLASTp based frequencies: 60.5% (invasive UPEC isolates) and 13.4% (non-invasive UPEC isolates).

Within CC69, 53 of 54 *papGII+* isolates formed a monophyletic clade and shared a conserved PAI. This suggests that this clade emerged after a single acquisition event involving a type II *papGII+* PAI in a common cladal ancestor (Fig. 5).

Similarly, within CC95, isolates of the *papGII+* sublineage shared a conserved type V PAI. Occurrence of the *papGII+* PAI at either the tRNA-*pheU* or tRNA-*pheV* site within one branch suggests possible excision and re-integration events within CC95 isolates (Supplementary Fig. 16). The population structure, supported by high-confidence bootstrap values, suggests that the PAI was present in the most recent common ancestor of all CC95 isolates but subsequently was lost in one branch (Fig. 5). Interestingly, this *papGII*-negative branch, which corresponds with so-called CC95 subgroup B[45], consisted mainly of non-invasive UPEC isolates and contained a sublineage characterized by integration of a *papGIII*-containing PAI at the tRNA-*leuX* site.

Within CC73, all four *papGII+* sublineages (here termed CC73-L1, -L2, -L3, and -L4) were part of a single large monophyletic branch characterized by the presence of *iuc* (Fig. 5). The occurrence of different *papGII+* PAI types within this *iuc*-containing branch suggests different *papGII+* PAI acquisition events within each sublineage. Alternatively, the different sublineages might have evolved from a common ancestor that contained multiple *papGII+* PAIs, such as seen in reference pyelonephritis isolate CFT073, which is phylogenetically similar to isolates of this branch and harbors distinct *papGII+* PAIs at its two tRNA-*phe* sites.

The four *papGII+* CC73 sublineages differed for the composition and insertion site(s) of their PAIs. Specifically, several isolates within sublineages CC73-L1 and CC73-L2, carried a type V PAI at the tRNA-*pheU* site, whereas others carried type III PAIs, including $PAI_{194-Pyelo-pheU}$ and $PAI_{US32-pheU}$, which seemingly resulted from recombination between a type V PAI and a *papGIII*-containing PAI (Supplementary Fig. 17). By contrast, isolates within sublineage CC73-L3 carried a type II PAI at the tRNA-*pheV* site, and those within sublineage CC73-L4 carried a type IV PAI. This PAI was integrated adjacent to a $PAI_{ABU83972-pheV}$-like island at the tRNA-*pheV* site.

Notably, isolates from the large *iuc*-containing branch within CC73 carried one of two *iuc* locus configurations, i.e., either (1) *shiF/iuc/iutA1*, on type II *papGII+* PAIs (found in CC73-L3), or (2) *shiF/iuc/iutA2*, on $PAI_{ABU83972-pheV}$-like islands (found in CC73-L1, CC73-L2, and CC73-L4). The latter was also found in isolates from sublineages associated with *papGII*-negative, non-invasive UPEC isolates, such as prototypic ABU strain ABU83972 and probiotic strain Nissle1917. These two *iuc*-containing PAIs might have been acquired independently or could have evolved from the same PAI, after its acquisition by a common CC73 ancestor.

**papGII+ sublineages within the pandemic UPEC lineage CC131.** The recently emerged lineage CC131 (dominated by ST131) is currently the leading cause of multi-drug-resistant *E. coli* UTI and bloodstream infections[12,46]. Although our main dataset comprises isolates collected over almost 40 years, it includes few CC131 isolates (n = 59), precluding a robust analysis of this lineage. Accordingly, to investigate the distribution of *papGII* within CC131 we combined these 59 CC131 isolates with 1017 additional publicly available CC131 isolates, which had, however, limited associated metadata (Supplementary Table 11, Supplementary Data 10). This analysis identified in all three main CC131 clades (A, B, and C) sublineages enriched with *papGII+* isolates (Supplementary Fig. 18). The largest *papGII+* fractions occurred within both subclade C2 (i.e., H30Rx; 216/521 isolates, 41%) and the *fimH27*-subclade within clade B (17/22 isolates, 77%).

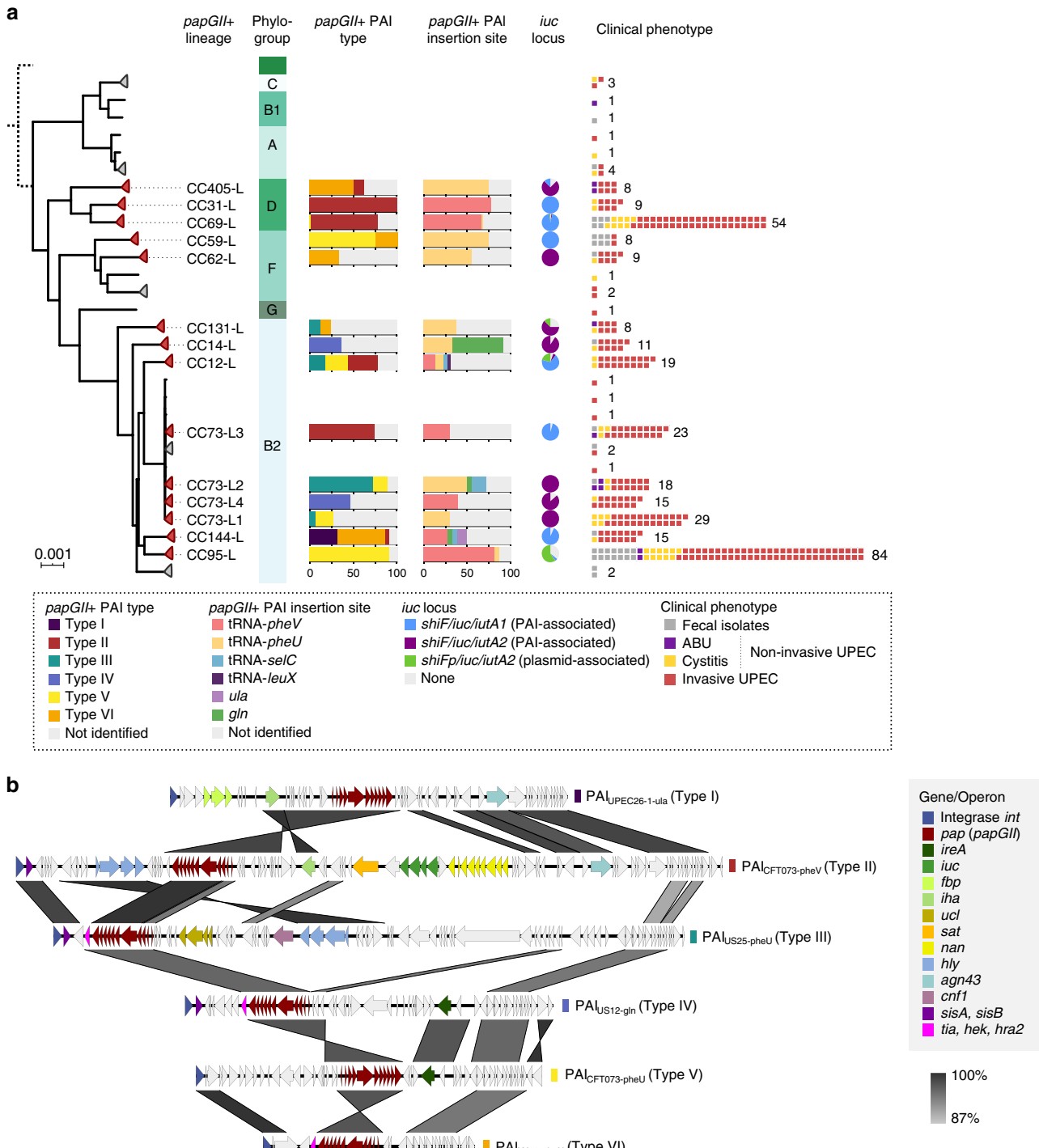

**Fig. 3 Genetic characterization of *papGII*+ *E. coli* lineages and *papGII*+PAIs. a** Maximum-likelihood phylogenetic tree of 333 *papGII*+ isolates based on 192,889 variable sites identified in a core genome alignment (2.573 Mbp). Phylogenetic lineages, defined by patristic distances, are collapsed on single nodes (indicated with triangles). Fourteen *papGII*+ lineages with >5 isolates (red triangles) were identified and named after their clonal complex (+L). Isolates in CC73 were investigated on an additional level of hierarchy and assigned to four *papGII*+ lineages (CC73-L1 to -L4) to account for the subclonal population structure with distinct characteristics. Each *papGII*+ lineage is labeled with the proportion of *papGII*+ pathogenicity island (PAI) types and insertion sites when their identification was possible. The presence of *papGII*+ PAI types was identified in complete or near-complete assemblies or predicted using a read-mapping-based approach. Fragmented assemblies, lack of resolved reference PAIs, or sequence deletions/insertions sometimes prevented the determination of the specific *papGII*+ PAI family type and insertion site (shown in gray). The proportion of isolates carrying PAI- or plasmid-associated *iuc* loci, the frequency of clinical phenotypes, and the total number of isolates are shown. The branch length of the outgroup (*papGII*-negative isolate 495_PUTI_Fec, clade I) was reduced (dashed line). The scale bar indicates the number of substitutions per site in the core genome alignment. A tree with expanded nodes and bootstrap values is shown in Supplementary Fig. 16. **b** Genetic organization of representative PAIs of the six identified *papGII*+ PAI types. The *papGII* operon, integrase gene, and virulence-associated genes are highlighted. The gradient scale shows the level of nucleotide identity. PAI sequences were compared and visualized using EasyFig[110]. The genetic organization of all 42 resolved *papGII*+ PAIs is shown in Supplementary Fig. 22.

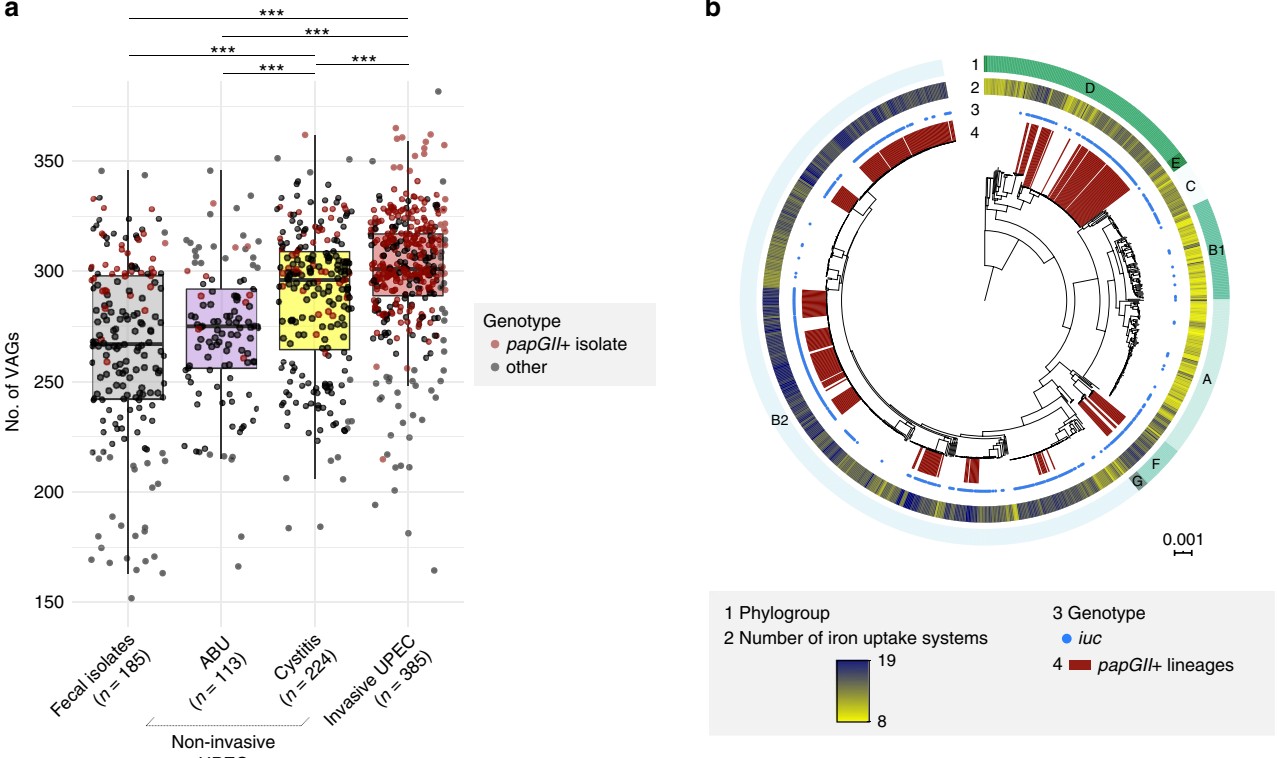

**Fig. 4 Number of virulence-associated genes (VAGs) and distribution of iron uptake systems. a** Boxplots showing the number of VAGs per isolate by clinical phenotype (fecal isolates, non-invasive UPEC isolates (asymptomatic bacteriuria (ABU), cystitis), and invasive UPEC isolates). *papGII*+ isolates are indicated as red dots. Asterisks indicate significant differences (***$P < 0.001$, two-sided Mann–Whitney U test, Bonferroni-corrected). Exact $P$ values are reported in Supplementary Fig. 13. Boxplot center lines: median; box limits: upper and lower quartiles; whiskers extend from the hinges to the highest and lowest values that are within 1.5×IQR of the hinges. Source data are provided in Supplementary Data 8. **b** Number of iron uptake systems (ring 2) per isolate and presence of *iuc* (ring 3) visualized on the phylogenetic tree. Twenty-two different systems involved in iron uptake were identified in our dataset, with 8–19 systems found per isolate. Phylogroups (ring 1) and isolates part of *papGII*+ lineages (ring 4) are labeled. The tree was visualized using iTOL[89].

Despite its recent emergence, CC131 appeared to have undergone multiple acquisitions of *papGII*+ PAIs. Among the 21 *papGII*+ CC131 isolates with complete or near-complete assemblies, 20 contained a type III and one a type IV *papGII*+ PAI (Supplementary Table 12, Supplementary Fig. 19). Among the remaining *papGII*+ CC131 isolates with available read data, a read-mapping-based analysis identified type III *papGII*+ PAIs in multiple isolates of clades A, B, and C, and, sporadically, type II, type IV, type V, and type VI *papGII*+ PAIs in isolates of clades B and C (Supplementary Fig. 18, Supplementary Data 12). Among the 192 CC131 UPEC isolates with available clinical phenotype data, *papGII* was significantly more prevalent among invasive UPEC isolates (67/133, 50%) as compared to non-invasive UPEC isolates (7/59, 12%: OR = 7.5, $P < 0.001$).

**Discussion**

In this study, we analyzed 907 *E. coli* isolates from individuals with ABU, cystitis, pyelonephritis, or urinary-source bacteremia, as well as from feces, thereby deriving a high-resolution population structure of UPEC associated with these different clinical contexts. The observed phylogenetic clustering of invasive UPEC isolates underlines the importance of the corresponding lineage-associated genetic determinants in UTI pathogenesis. Our large collection size provided sufficient statistical power to establish a genome-wide association of *papGII* with invasive disease, suggesting a causal relationship. Associations of most other genes were rejected by stringent statistical corrections for multiple testing and population structure.

The gene *papGII* has been associated epidemiologically with pyelonephritis and urinary-source bacteremia in directed, usually PCR-based studies[19–22,47], and was shown experimentally, with varying degrees of rigor, to contribute to kidney infection in murine and monkey models[48–50]. A key role for *papGII* in the pathogenesis of human pyelonephritis was confirmed recently by the finding that knock-in of *papGII* was sufficient to enable the *iuc*-positive but normally non-pathogenic *E. coli* strain ABU83972 (CC73, phylogroup B2) to cause pyelonephritis in humans[23]. In that study, PapGII was shown to enter kidney cells and to trigger renal tissue inflammation by reprogramming host gene expression[23]. These findings, together with ours, support a site (i.e., kidney)-specific and, hence, pathotype-defining role for *papGII*. However, our data also identify a comparatively rare locus, specifically *afaVIII*, as potential marker of invasive UPEC.

Apart from *papGII*, only the *iuc* locus exhibited a significant pan-genome-wide association with invasive vs. non-invasive UPEC. This suggests that iron scavenging, as mediated specifically by the aerobactin system (among multiple *E. coli* siderophores), is particularly important for tissue and bloodstream invasion during UTI. Abundant evidence supports both a critical role of iron uptake for UPEC and partial functional redundancy among the multiple *E. coli* iron acquisition systems[36]. Here, iron uptake was the virulence factor class most significantly enriched in invasive UPEC isolates, as compared to non-invasive UPEC or fecal isolates.

Previous studies identified *fyuA* (yersiniabactin siderophore) and *chuA* (heme-binding protein) as important pathogenicity determinants of ExPEC and UPEC[16,42]. Although here, after

adjustment for multiple comparisons and population structure, *fyuA* and *chuA* did not discriminate between invasive and non-invasive UPEC isolates, this does not exclude a role in general uropathogenicity for them. For example, we showed a clear association of *fyuA* and *chuA* with phylogroups B2, C, D, and F, which together harbored 94% of invasive UPEC isolates and 75% of non-invasive UPEC isolates. By contrast, *chuA* was absent and *fyuA* was uncommon (prevalence 20–35%) in commensal-associated phylogroups A and B1, suggesting that *fyuA* and *chuA* might play a role in urinary-tract colonization for both invasive and non-invasive UPEC. Future studies with specific datasets from fecal and urinary *E. coli* isolates could resolve the bacterial determinants of commensal vs. invasive and non-invasive UPEC.

Although we focused on the role of bacterial determinants in UTI pathogenesis, predisposing host factors also are critically important. Here, some of the putative invasive UPEC isolates that lacked recognized VAGs (including *papGII*) may have been misclassified based solely on clinical criteria, and actually caused their invasive infection due to host defense defects rather than heightened pathogenic potential. Debilitated hosts are often infected by *E. coli* strains with low intrinsic virulence, such as those from phylogroup A or B1 and that produce fewer virulence factors[51], and *papG* is less common among invasive isolates from compromised as compared with non-compromised patients[52].

Conversely, a substantial fraction of the present cystitis and ABU isolates, including strains that asymptomatically colonized the urinary tract over months[53], carried *papGII*, without causing invasive infection. Apart from possible stochastic effects, this may reflect in part the known variation in host susceptibility to UTI in relation to specific host genetic polymorphisms[54]; conceivably, these isolates' corresponding source hosts may have been more resistant to developing invasive UTI. In addition, the study's clinical phenotype assignments may have misrepresented some strains' virulence potential. For example, patients classified as having cystitis may have received prompt antibiotic treatment that prevented progression of *papGII*+ isolates to the kidneys, or their urine samples might have been collected before symptoms of pyelonephritis developed.

The observed preferential occurrence of *papGII*+ PAIs in expanded lineages associated with invasive UPEC is similar to observations of acquisitions of PAIs or plasmids linked to the clonal expansion of atypical EPEC[9] and ETEC lineages[10]. Whereas most ETEC and atypical EPEC lineages in those studies were from phylogroups A and B1, here all *papGII*+ lineages belonged to phylogroups B2, D, and F. Consistent with previous findings[4], our data demonstrate that isolates of phylogroups B2, C, D, and F carry greater numbers of VAGs than do isolates of other phylogroups, even in the absence of *papGII*+ PAIs, suggesting that such VAGs provide the genetic basis required to colonize extraintestinal sites such as the human urinary tract. This ability to colonize extraintestinal sites might be a precondition for the stable maintenance of acquired *papGII*+ PAIs and clonal expansion.

The spread of AMR among pathogenic *E. coli* poses a significant threat to public health. In particular, the emergence of *papGII*+ lineages within ST131 clades C1 and C2, which were shown to have high levels of AMR against commonly used antimicrobials to treat invasive UTI (ciprofloxacin, 3rd-generation cephalosporines, trimethoprim-sulfamethoxazole)[55,56], is concerning. In addition to bacterial virulence and host factors, AMR and associated treatment failure may impact the outcome of UTI, if not the patient's initial presentation. Here, invasiveness-associated sublineages were found among both multi-drug resistant clones (CC69 and CC131) and clones that typically exhibit broad antibiotic susceptibility (CC73, CC95)[12,13]. Regardless of

their usual AMR status, all such invasiveness sublineages shared the presence of *papGII* or *afaVIII*.

Considering the widespread presence of *papGII*+ PAIs and their stable maintenance in successful *E. coli* lineages, we speculate that PapGII confers a niche-specific selective advantage. P fimbriae with PapGII might contribute to urinary-tract colonization and thereby to pathogen transmission through shedding. Alternatively, PapGII might contribute to gut colonization; indeed, phylogroup B2 isolates that produce P fimbriae have been associated with increased persistence in the human intestine[57,58]. PapGII might also play a role in the colonization of zoonotic niches. Interestingly, *papGII* was found in up to 60% of avian pathogenic *E. coli* isolates from live or diseased poultry[59–61]. Human intestinal acquisition of UPEC/ExPEC has been associated with animal contact, consumption of high-risk food (including seafood, raw meat, or vegetables), and direct transmission between humans[13,62].

Previous studies demonstrated that UTI occurrence is linked to the relative abundance of the causal clone in the host's gut microbiota, and modulation of the gut microbiota has been suggested as a potential strategy to prevent UTI[63,64]. Here, *papGII*+ isolates from feces genetically resembled clinical *papGII*+ isolates, supporting the concept that the human gut is a reservoir of *papGII*+ isolates and that carriers might be predisposed to invasive UTI. In our dataset, *papGII*+ isolates (i.e., putative uropathogens) constituted ~20% of all fecal *E. coli* isolates.

Our study has notable limitations and strengths. Limitations include (i) an intrinsic sampling bias due to the variation in the distribution of *E. coli* clones, both over time and among different human populations; (ii) a potential underestimation of the contribution of rare variants or the combined effects of variants due to our stringent statistical adjustment by GWAS, leading to failure to detect true associations; (iii) inaccessibility of data on predisposing host factors, and (iv) reliance on observation/correlation, not experimentation. In addition, gene presence does not necessarily imply functional expression, and in vivo gene expression levels are unknown for the present study isolates. Notably, regulation of *pap* expression is complex[65,66]. Strengths include (i) the large number of study isolates with well-documented clinical phenotypes, (ii) the unbiased approach to identify genome-wide associations of genes and lineages with specific clinical syndromes, and (iii) use of a robust phylogeny in combination with long-read sequencing data to elucidate key evolutionary events.

In summary, our findings demonstrate that different UPEC lineages tend to behave as either invasive or non-invasive pathogens. Whereas a combination of multiple VAGs is likely required to colonize the urinary tract, *papGII* is specifically associated with invasiveness. A few *papGII*+ lineages account for most invasive UTIs across decades and continents, with recent or ongoing *papGII*+ PAI acquisition events likely leading to the emergence of new *papGII*+ lineages. These findings have implications for our understanding of the pathogenesis of invasive UPEC and, hence, for potential surveillance and control measures.

## Methods

**Bacterial isolates and genomes of the main dataset.** Genomes of 907 *E. coli* isolates from multiple collections were analyzed. These included 185 fecal isolates from human samples and 722 urinary-tract source extraintestinal isolates from human subjects with ABU (*n* = 113), cystitis (*n* = 224), pyelonephritis (*n* = 73), or urinary-source bacteremia (*n* = 312). Of these genomes, 738 originated from 13 publicly available collections (Supplementary Table 1). These were supplemented with genomes of 18 reference strains and 151 isolates that were sequenced as part of this study, including isolates from the collections LtABU, UZA_uro, MVAS-T_ABU, and MC_pye. Strain details are listed in Supplementary Data 1.

**LtABU collection.** The LtABU collection comprises isolates associated with long-term ABU obtained between 2017 and 2018 from non-catheterized residents

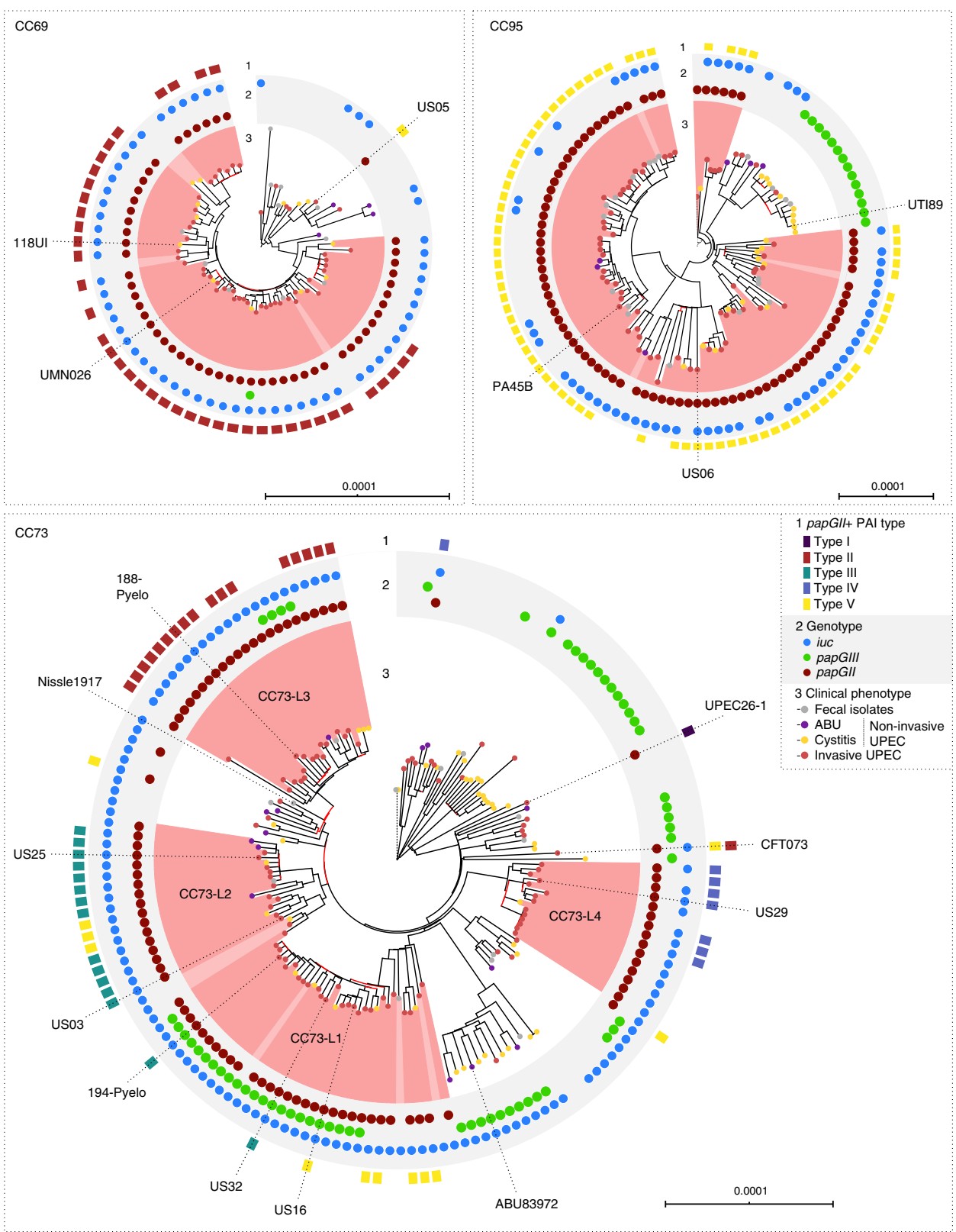

(>65 years of age) of two long-term care facilities (OCMW Destelbergen and WZC Immaculata Edegem) in Belgium. A total of 237 participants were screened for bacteriuria (>10⁵ CFU/ml) at up to four consecutive sampling timepoints over three months: at baseline and after ~2, 10, and 12 weeks. Bacterial species were identified using MALDI-TOF. The clonal relationship of consecutive *E. coli* bacteriuria isolates from the same patients was analyzed using random amplification of polymorphic DNA (RAPD) profiling. The RAPD assay consisted of two PCR reactions containing primer 1247 (5′-AAGAGCCCGT-3′) or 1283

(5′-GCGATCCCCA-3′), respectively, as described by Nielsen et al.[67]. Amplification was performed in 27 μL reactions with 1.5 μM primer, 0.2 mM dNTPs, 1.5 U SuperTaq polymerase (SphaeroQ), the supplied polymerase buffer, and 2 µl 100-fold diluted overnight culture. Cycling conditions were 15 min at 95 °C, 35 cycles of 1 min at 94 °C, 1 min at 38 °C (reaction primer 1247) or 36 °C (reaction primer 1283), and 2 min at 72 °C, followed by a final extension for 10 min at 72 °C. Amplification products were separated on 1.5% agarose gels in TBE buffer (1 h, 150 V), stained with GelRed (Biotium), and visualized under UV light (BioRad

**Fig. 5 Phylogenetic trees of isolates belonging to pandemic UPEC lineages CC69, CC95, and CC73.** Midpoint rooted maximum-likelihood phylogenies based on core genome alignments (CC69: 76 isolates, 25,753 variable sites in 4.006 Mbp core genome; CC95: 107 isolates, 17,674 variable sites in 4.054 Mbp core genome; CC73: 164 isolates, 25,939 variable sites in 3.857 Mbp core genome). Clinical phenotypes are labeled at the branch tips (ring 3). The presence of *papGII*, *papGIII*, and *iuc* is shown (ring 2). When identification was possible, *papGII*+ PAI types are labeled (ring 1). Fragmented assemblies, lack of resolved reference PAIs, and sequence deletions or insertions within PAIs sometimes prevented the determination of the specific *papGII*+ PAI type. Isolates part of *papGII*+ lineages are shaded in red; isolates part of the same lineages but lacking *papGII* in bright red. Isolates with complete or near-complete genomes used to investigate the genetic context of *papGII* are annotated. Red branch lines indicate nodes with bootstrap values <70. Branch lengths of distantly related isolates (outgroup) are reduced and indicated as dashed lines. The *papGII*-negative subclade in CC95 corresponds to the previously defined subgroup B (serotype O18:H7)[45]. The trees were visualized using iTOL[89].

ChemiDoc). Long-term ABU was defined here as at least three consecutive positive urine cultures covering a period of at least 10 weeks from participants without signs or symptoms of a urinary-tract infection and resulting in the isolation of clones with identical RAPD fingerprints. Forty-three long-term *E. coli* ABU isolates from 42 participants (median age 86, range 70–97) were identified and included in this study. One subject carried two *E. coli* clones (LtABU20, LtABU24) with distinct morphologies at three consecutive sampling points reaching maximum concentrations of >10^7 CFU/ml each. Most of the isolates originated from female participants (41/42) with incontinence (39/41). A comprehensive analysis of bacteriology and risk factors associated with ABU at one of the two collections sites was published previously[53].

**UZA_uro collection.** The UZA_uro collection comprises urosepsis isolates obtained from the Antwerp University Hospital UZA. Hospital records from 2015 to 2018 were screened for non-compromised patients with community-acquired sepsis who had positive urine and blood cultures with matching *E. coli* isolates (i.e., identical antibiotic susceptibility patterns) separated in time by ≤1 week. Immunocompromising conditions, surgery, pregnancy, urological intervention, and putative hospital-acquired infection were exclusion criteria. Sepsis symptoms were defined based on the SIRS criteria[68]. Infection was considered community-acquired when the sampling date of either blood or urine culture was no more than 2 days after admission to the hospital. Non-duplicate urosepsis isolates from 30 such patients (median 75.5 years, range 0–92) were available.

**MVAST_ABU collection.** The MVAST_ABU collection comprises asymptomatic bacteriuria isolates from male patients at the Minneapolis Veterans Affairs Medical Center (MVAMC) without long-term care facility residence in the previous year. The collection is a subset of isolates from various clinical sources described by Drekonja et al.[69].

**MC_pye collection.** The MC_pye collection comprises urine isolates from female patients (>18 years of age) with acute, uncomplicated pyelonephritis originally collected between 1994 and 1996 during a multi-center study in the United States[70,71]. Inclusion criteria were flank pain/tenderness, fever (>38 °C), and pyuria. Exclusion criteria included immunocompromised status, hospital admission, urologic abnormalities, and diabetes. Of the 70 isolates of the MC_pye collection, 39 were sequenced here and 31 had publicly available genomic data.

**Publicly available collections.** Publicly available genomic data for 875 isolates from 13 collections were considered for inclusion. Thirty-eight isolates were excluded due to low sequence quality (*n* = 25), inaccessible or corrupted read files (*n* = 8), inconclusive metadata (*n* = 4), or doubtful species (*n* = 1). Sixty-four isolates from the KTE_and Rec_cys collections were excluded due to duplicate sequencing of reported same-clone isolates from the same patients. The KTE collection originally comprised 48 urinary isolates from 47 cystitis psatients, 81 fecal isolates from the same patients, which partially (*n* = 40) corresponded to the infecting clone, and 67 fecal isolates from patients who never had a UTI (control). The 40 fecal isolates that were reported to match to the cystitis clone in the same patient[18] were excluded. The Rec_cys collection originally comprised 43 *E. coli* genomes obtained through consecutive sampling from 14 women with recurrent cystitis[43]. Putative same-clone isolates were excluded. Thirty-five isolates from the PUTI_cys, MC_pye, and MN_fec collections were excluded to correct for sampling bias. PUTI_cys, MC_pye, and MN_fec comprised isolates originally selected for sequencing based on presumptive ExPEC status (i.e., presence of at least two of the genes *papAH* and/or *papC*, *sfa/focDE*, *afa/draBC*, *iutA*, and *kpsMII*), with an ExPEC:non-ExPEC target ratio of approximately one[72]. For our analysis, strains were randomly sub-selected to reflect the unbiased ExPEC/non-ExPEC ratio of the source collection. Descriptions of all collections and details on exclusions are provided in Table 1, Supplementary Table 1, and Supplementary Note 1.

**Bacterial genomes of the CC131 dataset.** Assemblies from CC131 isolates of the main dataset were analyzed in the context of publicly available CC131 whole-genome sequences including assemblies of 92 isolates from children with febrile UTI[73] defined here as invasive UPEC isolates, 10 urosepsis isolates[17], 16 cystitis isolates[74], 52 isolates of various clinical phenotypes[46], and 799 isolates without

available metadata randomly downloaded from Enterobase[75]. Forty-eight complete or near-complete genomes of CC131 isolates identified on NCBI were also included in the analysis. Details of all isolates included in the CC131 dataset are provided in Supplementary Data 10 and collections are described in Supplementary Table 11.

**Short-read sequencing.** DNA was extracted from overnight liquid cultures (single colonies inoculated in 4 mL Mueller Hinton broth) using the MasterPure Purification Kit (Epicentre). Libraries were prepared using Nextera XT (Illumina), and sequencing performed on the Illumina MiSeq platform with 2 × 250 bp paired-end chemistries.

**Long-read sequencing.** Twenty-four isolates (Supplementary_Data 5) were additionally sequenced using single-molecule real-time (SMRT) technology (Pacific Biosciences). DNA was extracted using the MagAttract HMW DNA Kit (Qiagen) and sheared to ~8–10 kb using g-TUBE (Covaris). Libraries were prepared using the PacBio SMRTbell template preparation kit version 1.0 and pooled libraries of tagged isolates sequenced using the PacBio Sequel system.

**De novo assemblies and annotation.** Short-read data of 151 in-house sequenced isolates were pooled with raw reads downloaded from publicly available sources (NCBI's Sequence Read Archive or National Genomics Data Center, 724 isolates). Reads were trimmed using TrimGalore v0.4.4 (https://github.com/FelixKrueger/TrimGalore) and assembled de novo with Spades v3.13.0[76]. When long-read sequencing data were available, assemblies were obtained using HGAP 4[77] with default settings with subsequent short-read polishing with Pilon implemented in Unicycler v0.4.8[78] and compared to hybrid assemblies performed with Unicycler v0.4.8 using default settings. The best assembly based on the number of contigs and N50 was used for further analyses. HGAP assemblies, which usually did not recover plasmid sequences, were combined with plasmid assemblies obtained from hybrid assemblies in Unicycler (see Supplementary Data 6). Contigs representing plasmids were predicted using MLPlasmids[79] and replicons identified using the PlasmidFinder database[80] in ABRicate v0.9.3 (https://github.com/tseemann/abricate) (minimum sequence coverage/identity 70/90%). Pre-assembled data were downloaded from NCBI when raw read data were not accessible (17 reference strains and 15 isolates of collection Rec_cys, see Supplementary Data 1). Assembly quality was assessed using Quast[81]. Genomes of all 907 isolates were annotated using Prokka v1.13.3[82].

**Phylogenetic analysis.** To generate core genome alignments, collinear blocks (i.e., homologous genomic regions free from internal rearrangements) in assemblies were identified and aligned using Parsnp v1.2[83]. Separate core genome alignments were generated for the main dataset (907 isolates, reference genome UMN026, GCA_000026325.2) and isolates belonging to CC69 (76 isolates, reference genome UMN026, GCA_000026325.2), CC73 (164 isolates, reference genome CFT073, GCA_000007445.1), CC95 (107 isolates, reference genome UTI89, GCA_000013265.1), and CC131 (1076 isolates, reference genome JJ1886, GCA_000493755.1). Prophage regions in reference genomes were identified by PHASTER[84] and masked from the respective alignments. Core genome alignments were used to construct maximum-likelihood (ML) trees using RAxML v8.2.12 with the generalized time-reversible (GTR) model and gamma distribution[85]. One hundred bootstrap replicates were performed to assess support for the phylogeny. ClonalFrameML v1.12[86] was used to account for recombination events and to correct branch lengths with the best RAxML tree as starting tree. To identify lineages enriched with invasive UPEC isolates, the pre-computed phylogeny of all 907 isolates or of CC69, CC73, or CC95 isolates was partitioned into clusters on multiple levels of resolution using Bayesian hierarchical clustering implemented in the R package fastbaps[87]. Clusters identified in different phylogenies were merged and the 129 clusters were tested for significant phenotype enrichment among UPEC isolates using Fisher's exact test. *papGII*+ lineages were defined based on patristic distances (cutoff 0.00032) in the maximum-likelihood tree of *papGII*+ isolates using RAMI[88]. A lower cutoff distance (0.000052) was chosen for isolates of CC73 to address the heterogenous population structure within this CC and nested *papGII*+ clades combined. Phylogenetic trees and isolate metadata were visualized in iTOL[89] and annotated using Inkscape 0.92.

**Genome-wide association studies**. Clusters of orthologous genes (COGs) representing homologous genes with shared sequences were identified using Roary v3.12.0 with a BLASTp identity cutoff of 95%[90]. The pan-genome of all isolates comprised 37,717 COGs, of which 3067 COGs represented the core genome (present in ≥99% of all isolates). Pan-genome-wide association studies (pan-GWAS) on the COG presence-absence matrix were performed on 30705 COGs (of which 2237 were present in all isolates) identified in 722 UPEC isolates using the R package treeWAS v1.0 with Bonferroni correction[91]. Fecal isolates were excluded from pan-GWAS and DBGWAS analyses due to their unknown urinary phenotype. Associations of each COG with invasive or non-invasive UPEC isolates were calculated using the Fisher's exact tests. The significance threshold (raw $P = 1.42 \times 10^{-18}$, corresponding to a Bonferroni-corrected $P = 4.04 \times 10^{-14}$) was determined from a simulated dataset accounting for population structure as described in treeWAS. Manhattan plots were generated using the R package CMplot (https://github.com/YinLiLin/R-CMplot). PAIs in the genome of reference strain UMN026 were identified using Islander implemented in IslandViewer 4[92]. Genetic associations were additionally analyzed with DBGWAS v0.5.4[26] with a $q$-value threshold of 0.05.

**Genotyping**. Phylogroups were determined according to phylogenetic clustering and supported by in silico typing using ClermonTyping v1.4[93]. Multi-locus sequence types (ST) of the Achtmann scheme were determined using srst2 v0.2.0[94] with default settings or mlst v2.16.1 (https://github.com/tseemann/mlst). Identified STs were grouped into clonal complexes (CC) according to the scheme available at EnteroBase[75]. Alleles of type 1 fimbrial tip adhesin gene *fimH* were identified using FimTyper v1.1[95]. H and O serotypes were predicted in silico with the EcOH database[96] using srst2 v0.2.0[94] with default settings, or, in case no raw read data were available, ABRicate v0.9.3 (minimum sequence coverage/identity 70/90%). Pointfinder[97] was used to search for mutations in the quinolone resistance-determining regions (QRDR). Resistance genes were identified using ABRicate v0.9.3 in conjunction with the resfinder database[98] (minimum sequence coverage/identity 70/90%). Prophage genes and regions were identified using PHASTER[84] with annotated genbank files as input. Clades of CC131 were confirmed using ABRicate v0.9.3 by screening assemblies against the clade B-specific allele of *prfC* identified in isolate KTE6 (minimum sequence coverage/identity 100/99%), the clade C2-specific allele of *ybbW* identified in isolate JJ1886 (minimum sequence coverage/identity 100/99%), and the clade C1-specific allele of a gene with locus tag U12A_RS05235 identified in isolate U12A (exact matches).

**Screening for virulence-associated genes**. An *E. coli* virulence-associated gene database (EcVGDB)[99] was compiled from two separate collections of *E. coli* virulence factors, ecoli_VF_collection[100] and ecoli_vf (https://github.com/phac-nml/ecoli_vf), which are both based on the virulence factor database VFDB[101]. Redundant sequences identified using cdhit[102] were removed. The database was supplemented with additional virulence-associated genes (VAGs) from the literature, resulting in 1368 sequences representing 1072 VAGs of 14 virulence factor categories. Genome assemblies were screened against the EcVGDB using ABRicate (minimum sequence coverage/identity 70/90%). The presence of alleles of the virulence gene families *papG* and *afa/dra* was additionally assessed by read mapping using srst2[94] using default settings, accepting hits with no or minor mismatches (SNPs or indels). *iuc* locus variants were determined by mapping reads to the reference genes *shiF*, *shiFp*, *iutA1*, and *iutA2* obtained from the genomes of isolates CFT073, US06, and IAI39 (srst2[94], default settings). Hits with indels or holes were rejected. When read data was not available, *iuc* variants were determined using ABRicate with the three complete operons as reference (minimum sequence coverage/identity 70/90%).

**Identification and clustering of mobile genetic elements**. Completely resolved *papGII*-containing PAIs, and PAIs or plasmids containing the *iuc* locus, were extracted from high-quality genome assemblies. To estimate their similarity, mash distance matrices were produced using mashtree v1.12[103], and the genetic elements hierarchically clustered in R v3.5.3. *papGII*-containing PAIs were clustered by applying a mash distance cutoff of 0.04. Mash distances are an estimate of sequence similarity calculated from the fraction of shared k-mers (Jaccard index) in MinHash sketches[104]. PyANI v0.2.9[105] with BLAST+[106] was used to identify the pairwise alignment coverage of PAIs. For the comparison of resolved PAIs of the same type across clonal complexes, the lower of the two alignment coverage values resulting from subject and query choice of each pair was chosen. To predict *papGII*+ PAI types in *papGII*+ isolates with insufficient sequence assemblies, their sequencing reads were mapped to the 42 resolved *papGII*+ PAIs using srst2[94] with default settings. Identified hits with ≥90% identity and ≥90% coverage were screened for homologs with no or minor mismatches (SNPs or indels) and assigned to the corresponding *papGII*+ PAI type. Hits with large deletions or truncations ("holes") were counted as mismatches. Hits in isolates with both *papGII*+ and *papGIII*-containing PAIs, particularly observed in *papGII*+ lineages of CC73, could lead to ambiguous hits which were excluded due to possible recombination events between the two *pap*-containing PAIs.

**Detection of integration sites of *papGII*-containing PAIs**. ABRicate v0.9.3 (minimum sequence coverage/identity 70/90%) was used to query the genomic position of *papGII* and housekeeping genes surrounding the six identified integration sites (tRNA-*pheV*: *speC* – *kpsFII* or *gspM/yghD*; tRNA-*pheU*: *yjdC* – *cadC*; tRNA-*selC*: *yicL* – *yicJ*; tRNA-*leuX*: *yjgB* – *gntP/uxuA*; *ula*: *ulaE* – *ulaD*; *gln*: *glnP* – *glnH*) in each genome assembly. Integration sites were then inferred from the chromosomal distance between *papGII* and the surrounding housekeeping genes. The integration site could not be inferred from assemblies with highly fragmented PAIs, which was often the case in isolates with multiple *pap* operons.

**Multiple sequence alignments and visualization of genomes and PAIs**. Gene and protein sequences were aligned using Muscle v3.8.31[107] and their phylogenetic relationship reconstructed using a GTR model in Mega-X v10.0.5[108]. Percent sequence identity matrices were calculated using Clustal2.1[109]. Schematic representations and comparisons of genomes and PAIs were generated using EasyFig 2.2.3[110].

**Statistical tests**. Statistical analyses were performed using R v3.5.3. Frequency counts were compared using two-tailed Fisher's exact test, while continuous variables were analyzed using two-tailed Mann–Whitney U test. Bonferroni adjusted $P$ values of <0.05 were considered to reflect statistical significance, except in the GWAS analyses.

**Material availability**. Requests for obtaining clinical isolates collected as part of this study should be addressed to the corresponding author. Exchange of clinical isolates should always be in agreement with the University of Antwerp.

**Ethics**. Ethical approval for the study was received from the ethics committee UZA (approval No. 17/08/081, No. 20/11/119, and No. 18/10/122). Written informed consent was obtained from all subjects participating in the clinical investigation or, when the participant was not capable of giving consent, by his or her legal representative.

**Reporting summary**. Further information on research design is available in the Nature Research Reporting Summary linked to this article.

## Data availability

Illumina and PacBio reads generated for this study are available at the NCBI Sequence Read Archive (SRA) under BioProject no. PRJNA592372. Complete or draft genome assemblies have been submitted to NCBI GenBank. Individual accession numbers are provided in Supplementary Data 1. Sequences of *papGII*+ PAIs and the curated *E. coli* virulence gene database (EcVGDB) are provided at https://github.com/MBiggel/UPEC_study (https://doi.org/10.5281/zenodo.4079473). An interactive version of the core genome phylogeny of the 907 *E. coli* isolates is accessible at https://microreact.org/project/O4QAYAJWw. All other relevant data are available from the corresponding authors. Public data utilized in this study include genomic data (accession numbers provided in Supplementary Data 1 and 10), the databases plasmidfinder, resfinder, EcOH, ecoli_vf, ecoli_VF_collection, and the EnteroBase ST/CC scheme.

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

## Acknowledgements
We want to thank all participants who donated samples for this study; the staff of the long-term care facilities OCMW Destelbergen and WZC Immaculata Edegem, in particular Marleen Paelinck, Wouter Rogiest, and Hans Van Braeckel; Stefan Heytens and Katrien Latour for help in organizing the sample collections and reviewing patient data; staff of the Laboratory of Medical Microbiology (LMM) for assisting in genome sequencing; Julien Coussement and Brian Johnston for sharing bacterial isolates or sequencing data; and all scientists who deposited their sequencing data on publicly available archives, which has made this study possible. This publication made use of the PubMLST website (http://pubmlst.org/) developed by Keith Jolley[111] and sited at the University of Oxford. The development of that website was funded by the Wellcome Trust. The computational resources and services used in this work were provided by the VSC (Flemish Supercomputer Center), funded by the Research Foundation—Flanders (FWO) and the Flemish Government—department EWI. The work has received funding from the European Union's Horizon 2020 research and innovation program under the Marie Skłodowska-Curie grant agreement No. 675412. This work is also supported in part by Office of Research and Development, Department of Veterans Affairs (U.S.).

## Author contributions
M.B., P.M., and S.V.P. designed the study. M.B. collected the samples and performed the experimental analyses. V.M. and J.J. contributed to isolate collections. M.B. performed the bioinformatics analyses with input from S.V.P. and P.M. M.B, S.V.P., and P.M. wrote the manuscript. This was revised by B.X., H.G., J.J., K.N., and N.F.-M. All authors have read and approved the manuscript.

## Competing interests
The authors declare no competing interests.
