## [Peer Review File · Nature Communications]

Reviewers' Comments:

Reviewer #1:

Remarks to the Author:

Review for Nature Communications

Title: Horizontally acquired papGII-containing pathogenicity islands underlie the emergence of invasive uropathogenic Escherichia coli lineages

Comments for Author

The authors have conducted a study with the aim of constructing a high-resolution population structure of uropathogenic strains (UPEC) including samples collected from individuals with different clinical manifestations. In addition, the authors searched for genetic markers that are associated with a specific clinical phenotype, focusing in invasive UPEC.

UPEC are difficult to differentiate from non-pathogenic E. coli and that is an important aspect. If it was possible to, in an early stage, identify what "type" of E. coli is causing the infection (UTI, pyelonephritis etc.), for example if it was an E. coli with specific virulence factors that leads to an invasive infection along with antibiotic resistance profiling it would help in choice of treatment. In the manuscript the majority of the E. coli isolates are collected from patients with an infection, however regarding the faecal samples I am missing information on how these samples were classified as UPEC. I would appreciate if you could clarify that because that might explain why they were included in the first place. This is somewhat explained in the methods section but I would suggest you add a sentence on this in the results section.

The major finding was that a specific adhesin, papGII, was identified on pathogenicity islands present in a sub-lineages and was identified as a key marker for lineages encompassing isolates collected from patients with severe symptoms, i.e. invasive UPEC. Furthermore, the authors performed a genome-wide association analysis identifying papGII being associated specifically with invasive UPEC. Suggesting that the presence of papGII could be an invasive UPEC-specific determinant. This genetic marker could be used to early on identify the potential if a urinary tract infection could become invasive or not. The findings are important for the UPEC field, especially as UPECs are difficult to define and the proposed work presents VAGs associated with invasive UPEC. Iron sequestering systems is a common trait among UPEC strains, and this study has identified one specific siderophore, iuc, also associated with invasive UPEC. Hopefully, additional UPEC associated VAGs or other genetic markers that are related to UPEC isolates will be identified if similar analysis could be conducted on a larger dataset. The findings will not only be of importance for the UPEC field but in the field of E. coli research in general, e.g. specifically how E. coli pathotypes evolved and acquire gene(s) needed for its pathogenicity and survival in different environments.

The conclusion made in the manuscript are based on multiple different analysis. For example, the population structure, differentiating between isolates that are invasive vs non-invasive and the identification of papGIII being enriched in the sub-lineages that mainly encompass invasive UPEC. Furthermore, the genomic markers identified as being associated with invasive UPEC was further confirmed with an alternate GWAS method strengthening their findings. The authors also explain both weaknesses and strengths with the dataset and the analyses that has been executed.

The article is well-written and the figures are detailed, yet easily understood. The genome comparisons and PAI comparisons figures were generated using Easyfig. Could you add that in the legend of the figure. The same for the figures with phylogenetic trees (using a specific tool, in R or what have you). Furthermore, the manuscript is extensive with seven figures, maybe consider choosing 3-4 figures to be included in the main article. Maybe you can combine figures or just choose to add more figures in the supplementary section.

The work is extensive and well executed and I fully support the manuscript for publication with minor revision. I have additional comments that you can find in the attached pdfs.

Reviewer #2:
Remarks to the Author:
General Statement

Overall, this was an interesting paper on the association of the putative virulence factor papGII with invasive UPEC lineages. The authors have undertaken a lot of genomic based analyses, however, I can't recommend that it is accepted in its current form. This manuscript requires significant reworking to improve clarity of the key findings as at times it was difficult to follow and understand the significance of the work.

Major Comments

1) The introduction didn't background all the key concepts and importantly, the rationale for the study was unclear from the introduction. This is necessary, particularly for Nature Comms which has a broad readership, and there can't be an assumption of knowledge.

- A key area not introduced adequately given the focus of the paper is the role of the pap operons and the role it potentially plays in host/pathogen interactions and why it is considered a putative virulence factor for ExPEC. At present there are only 5 lines in the introduction. As the manuscript currently reads – it is unclear at the end of the introduction why the papGII is the focus of this research.
- There is some text in the discussion which would be better in the introduction (lines 293-299)
- Further, the concept of pathogenicity islands and the role they play in virulence in bacteria is also not clearly addressed. This directly relates to the acquisition of putative virulence factors associated with UPEC and is important to background.
- Clinical terms such as pyelonephritis and cystitis need to be clearly defined as what these terms refer to. At present this reads as assumed knowledge which it can't be.
- The pathotype terms UPEC and ExPEC are used interchangeably through the introduction. This is incorrect. While all UPEC are ExPEC, not all ExPEC are UPEC. Please ensure these terms are defined and used appropriately.
- No mention is made of antimicrobial resistance in the introduction, which given AMR is addressed in the results and discussion, this needs to be briefly addressed.
- At the end of the introduction I was unclear on the specific gap in knowledge the authors were addressing and what the key research questions were for this study.

2. The study design was not clearly explained (both in the methods and the results).

It was unclear why the 16 genome datasets (4 novel ones in this study and 12 others) were appropriate to address the research question.

- Are there sampling biases that need to be addressed? E.g. one of the new datasets MVAST_ABU was a subset of a larger study – how were these isolates refined? In the public data - line 427 states 35 isolates were excluded for sampling bias?
- Most of the data in this study is public data. It is great to see public data used in this way, however further detail of why these data were chosen and not others is required. What were the selection parameters for including these public data and how might it impact upon the study design?
- The inclusion of a table in the main body of the manuscript may be useful to the reader to understand where and when the data has been collected. This could include study, number of isolates, country of isolation, timespan of samples, clinical presentation associated with the samples, age of the patients (looks as though most were >60 years), and refs to the studies for the public data. There are some figures in the supplementary that try to address but I think consolidating into one table may help
- No details were provided on the MC_Pye study in the methods.

3. Inclusion of the ~180 faecal isolates. I am unclear on what these isolates bring to the study and also the sampling strategy for their inclusion. They are already excluded from the GWAS analyses in this study. At times they are used as comparative reference for rates of genes present/absence in invasive v non-invasive UPEC. However, it doesn't appear these faecal associated isolates were selected to be broadly representative of the diversity of E. coli. These data could be removed from the study which would help to simplify the study or their inclusion needs to be appropriately addressed.

4. Long read assemblies – further detail is required

- How/why were isolates selected for long read sequencing?
- Were default settings used in all cases for both HGAP and Unicycler?
- Was there any selection for long reads based on length?
- The authors note that plasmids were lost from HGAP approaches that were obtained from Unicycler. Please provide details of the long read assemblies so the reader can tell what assembler was used to generate the data. This could be provided in a supp table with info for each chromosome and plasmid in each isolate in addition to accession numbers for the complete genomes generated in this study.

5. Screening for genes in assemblies and short read data. Please state if default parameters were used or if changed to ensure reproducibility.

6. Phylogenetic analysis. Please give extra details about what references were used. Were different alignments made for each CC specific analysis? Line 479 states reference genomes had phage regions masked. Reference name and accession would address this.

7. I have some concerns about how CC73 was split into four sub-lineages by using a patristic distance in RAMI that was different to that used for everything else. What were the grounds for this lineage being further split?

8. Manhattan plots – Fig 2 and Supp Fig 4. These were confusing to interpret initially as they have the same figure title and same figure legend until the second half there the difference in what is being shown is stated. For Fig 2 – this needs to be clear that only showing data for 4764 COGs that are part of the reference genome in the title. The same applies for Supp Fig 4 but in reverse which shows the remaining 25941 COGs.

9. Characterisation of PAIs. A total of six PAIs with papGII are characterised in this study from the 35 complete genomes. More analyses in characterising the PAIs would enhance one of the main findings of this study.

- Fig 3 – in the 14 lineages with papGII – not all lineages are shown as investigated for PAI. Why not? From the figure it appears that ~50% of PAI in the 14 main lineages were characterised. This significantly limits what you can infer about PAI acquisition and maintenance in the different lineages.
- Fig 3 - For the papGII lineages where no PAI were reported and those such as CC73 where large number of isolates are shown in grey meaning not determined – were these investigated further? Is it possible to try and characterise these novel lineages from the assembly data – at least of the immediate papGII region? E.g. in CC14 the tRNA site was identified but was a novel variant. Is it possible to characterise these PAIs?
- The nomenclature of having the tRNA site included in the name of the PAI with papGII is difficult – especially where integration was at a different tRNA site. E.g. CFT073-pheU but in fig 3 the most common site was pheV. Suggest using -1, -2 etc to delineate different PAI with papGII from same genome.

10. Fig 4 – PAIs and VFs.

Difficult to see the detail on panels B and D. Most of the info in panel C is in Fig 3. Could consider a restructure to highlight key results.

How does the six families of PAIs related to the core genome structure?

11. Distribution of virulence factors (VFs). Unsure of distribution of VFs by CC. The authors looked at the virulence factors by the six main PAIS except it looked like these only include ~50% isolates. Were different CCs characterised by different complements of VFs? How might this impact clinical outcome (particularly as not all papGII positive isolates were invasive UPEC)? The iuc gene is explored in a little detail but only in three CCs. Also – line 185 states 'significantly more VAGS' – this implies statistically significant – correct?

12. Three pandemic ExPEC lineages.

Why the switch in the language to ExPEC when been using UPEC?

Colours in Fig 6 – please use different colours – e.g. grey is used for both iuc presence and faecal isolates. PAI and cystitis are very similar shades of yellow. Also what is the significance of pink shading (not the red for papGII)?

13. Clarity of language and overstatement of results. The language also needs to be toned down as no lab based experiments to look at phenotype of the invasive isolates were undertaken and caution needs to be taken when inferring putative virulence from genomic data alone. For example.

- Role of afa - Line 169 – states afa has a similar role in (putative) pathogenesis of papGII. What is the basis for this claim? Could be a hypothesis (for discussion) but the results here don't show this. This is repeated in the discussion at lines 299-301.
- CC95 – lines 233-234 – states there has been excision and reintegration of the PAI. What was the supporting evidence for this? Supp Fig didn't address how could make this statement for the CC.
- CC73 – iuc gene – alternative hypothesis is that the iuc gene was acquired and then largely maintained within the CC73 lineage with different PAIs then acquired. Unclear how different the sub-lineages of CC73 were – both CC73-L1 and CC73-L4 have <50% of the isolates characterised for PAIs. Further - was any investigation undertaken of the genomic context of iuc? Was this the same in CC73 from different sub-lineages or different? Potentially acquired once and then lost in a handful of isolates?
- papGII in non-invasive isolates – lines 331 – due to host being 'relatively resistant' to invasive UTIs. What is the data to support this statement and what does 'relatively resistant' mean?
- Further, no data are presented on the papGII interacting with host cells. While data presented suggests papGII may be associated with the invasive phenotype, and a previously study has shown some kind of functional role of papGII, it is worth noting that this study only presents genomic data. The functionality of papGII in the different CC backgrounds has not been shown, much less that this is the definitive pathogenesis mechanism e.g. lines 120-121 - 'specifically genetically determined pathogenesis mechanism' for invasive UPEC.

14. Structure of results. The results section felt disjointed and it was difficult to follow at times. Suggest a restructure of the results so clear from each section what the main findings were and this would be improved by having full paragraphs rather than a series of very short blocks of text. This would greatly improve the readability of the manuscript.
e.g. results on distribution of papGII is split over 9 paragraphs – the shortest on being 2 sentences and 4 lines (lines 171-174).

15. Future work – this study presented some interesting data which could form the basis for future genomic and wet lab based experiments. Any considerations of what would be important research questions rising from these data?

16. Use of supplementary tables / figures. Please ensure there are referred to in order in the manuscript. For the tables went from supp table 1 to supp tables 9 and 10 and with supp figures from 1 and 2 to supp fig 5.

Minor Comments

- Please check language for terms such as 'non-demented male patients' (line 419). I was unsure what this meant. If this is a medical term it needs a clear definition.
- Great to see the data on microreact so ppl can explore it interactively. A minor point is that some of the countries from Korea, France, Germany, Australia and Mexico aren't shown on the map as the latitude and longitude data are included in table to better showcase the diversity of countries included in this study
- Fig 4 – please label X axis for barplots – with proportion 0, 25, 50 75 and 100.

Reviewer #3:

Remarks to the Author:

General comments: This report describes phylogenomic analysis of 722 uropathogenic *E. coli* (UPEC) isolates from different clinical sources (asymptomatic bacteriuria, cystitis, pyelonephritis, and urosepsis) and found papGII to be significantly associated with invasive disease (pyelonephritis, urosepsis). The report suggests invasive UPEC lineages emerged through repeated horizontal acquisition of diverse papGII-containing pathogenicity islands. This study addresses an important and unresolved question regarding UPEC virulence factors specifically associated with invasive UTI-causing *E. coli*. Using a large collection of UPEC isolates from a variety of collections, the investigators applied pan-genome-wide association analysis and narrowed the previously-reported invasive virulence-associated genes (VAGs) to just papGII and iuc loci. Further analysis confirmed papGII to be strongly associated with the invasive UPEC lineages.

The major limitation of this study, however, is the lack of reliable host-related data as well as clinical management data, which are important determinants of invasive UTI. The study lacks detailed consideration (e.g., multivariate analysis) of other relevant factors that contribute to invasive disease and attributable fraction each of these factors (including papGII-UPEC) to invasive UTI.

Specific comments:

1. The clinical diagnosis of pyelonephritis is highly subjective and its use in the definition of "invasive" disease may be unreliable.
2. The authors acknowledge that the absence of well-documented information about host-related predisposing medical conditions may have caused misclassification of invasive UPEC strains. In the Methods, the authors state that for the urosepsis isolates, they excluded isolates from patients who were immunocompromised, pregnant, had surgery or urological intervention, and had putative hospital-acquired infection. However, they also state that the median age of the patients from which the urosepsis isolates were analyzed was 75.5 years. Most cystitis cases would be expected to occur in a much younger age group. Age itself, regardless of predisposing conditions, is a major determinant of invasive disease. What was the median age of the patients who had non-invasive UTI? Without controlling for age, it is misleading to attribute papGII to so-called "invasive UPEC".
3. Same for gender. Cystitis is much more frequent in women while invasive disease is similarly distributed in both genders, especially older patients. Was prostatitis included in the definition of invasive UTI?
4. The study does not describe the potential association of any of the antimicrobial drug-resistance (AMR) genes or a set of such genes with the two clinical phenotypes. Infections caused by UPEC strains containing AMR genes conferring resistance to broad-spectrum antibiotics (e.g., ESBL-producers) or strains containing multiple AMR genes lead to greater morbidity and complications due to incorrect drugs or delayed initiation of a correct treatment regimen.
5. In the Discussion, the authors speculate that, in addition to its role in invasive pathogenicity, this papGII may be involved in transmission via clonal expansion in animal niches other than the human

host, such as poultry. Since ExPECs are frequently isolated from poultry, it would be interesting to know if papGII is overly represented in such isolates.

6. Table 1 shows % frequency of papGII in invasive vs non-invasive UPEC isolates. It would be more informative to show numbers also (number papGII/total).

In summary, while this study makes an interesting observation, the interpretation of the observation is somewhat pedestrian. The analysis presented in the report is limited by the lack of consideration of many other factors that affect the definition of "invasive" disease. In fact, the so-called invasive disease is a multifactorial clinical entity affected by host-related and clinical management factors. These factors may create distinct host conditions or niches that favor selection of UPEC strains with certain characteristics. Strains that carry papGII may just be selected in these distinct host-related niches and may have little to do with their intrinsic invasive properties.

REVIEWER COMMENTS

We thank the reviewers for their in-depth reading and constructive suggestions to improve our manuscript. The adaptations made to the manuscript are listed below, referring to the line numbers (L)

Reviewer #1 (Remarks to the Author):

The majority of the E. coli isolates are collected from patients with an infection, however regarding the faecal samples I am missing information on how these samples were classified as UPEC. I would appreciate if you could clarify that because that might explain why they were included in the first place. This is somewhat explained in the methods section but I would suggest you add a sentence on this in the results section.

- Fecal isolates were not associated with intestinal or extraintestinal infections and included as context isolates for genomics analyses (added to introduction L 85 and results L 97)
- Supplementary Note 1 on “Inclusion criteria for isolate collections” was added.
- The fecal isolates were not included for GWAS between invasive versus non-invasive UPEC, but included in comparisons: we described the distribution of fecal isolates in the phylogroups in comparison to UPEC isolates (L 108 and Supplementary Note 2), estimated the prevalence of *papGII* in fecal isolates (L 130) and described virulence gene profiles (L 235, Fig. 4a).

Genome comparisons and PAI comparisons figures were generated using Easyfig. Could you add that in the legend of the figure. The same for the figures with phylogenetic trees (using a specific tool, in R or what have you).

- The tools used to visualize genome comparisons and phylogenies were added to the respective figure legends.

Furthermore, the manuscript is extensive with seven figures, maybe consider choosing 3-4 figures to be included in the main article. Maybe you can combine figures or just choose to add more figures in the supplementary section.

- We reduced the number of figures from 7 to 5.
- Fig. 7 was moved from the main text to the Supplementary Material.
- Fig. 3 and 4a were combined and Fig. 4b, 4c, and 4d were moved to the Supplementary Material.

Reviewer #1, minor comments in pdf file

How did you determine that they were in fact UPEC isolates? Based on a set of VAGs that were present?

- UPEC isolates are defined based on the clinical phenotype, irrespective of their genotype. This was clarified in L 96 and Supplementary Note 1.

I am a bit confused to the number of samples that was included in the genomic analysis. In the first section of the methods 907 samples were included in the genomic analysis. In total, 722 upec + 185 fecal samples + 151 additional isolates = 1058 samples.

- The main dataset comprises 907 isolates (722 UPEC + 185 fecal). Of those, 151 isolates were sequenced as part of this study; 756 isolates were publicly available. This was clarified in L 96 and L 102.

The reason for including the fecal samples have not been described. I think it is important you describe why they were included as well as referencing to what type of samples these were.

- Explained in Supplementary Note 1 (see also first comment).

“Implications for the early detection and prevention of disease”; This is not brought up in the discussion, I am curious to how your findings could help with prevention of disease.

- *papGII*-lineage specific antigens could serve as valuable immunization targets. For example, O-antigens of *papGII*+ lineages could be considered in the ExPECV10 vaccine, which is currently in development. However, to keep the main message of our manuscript focused, we did not elaborate on applications of our findings for prevention and “prevention” was removed from the abstract (L 28).

There is a phylogroup G as well.

- The recently defined phylogroup G was added to the analyses and Figures.

How do you define what a VAG is? In your excel file with VAG profiles you are including many different vir factors which most likely are more specific in other *E. coli* pathotypes

- There is no specific VAG database for UPEC or ExPEC available. We used an *E. coli* wide approach and compiled our own database based on available curated *E. coli* virulence gene databases. (Methods section L 623 and Supplementary Data 8).

I suggest you also add the ST profile of the isolates as many publications on *E. coli* refer to ST.

- The dominant ST profiles were added to Fig. 1 and, for interactive exploration, to the tree on microreact (<https://microreact.org/project/O4QAYAJWw>).
- We refer to the STs in the results section on L 110.
- STs of all isolates are reported in Supplementary Data 1.

The *papGII* gene has been mentioned above with out an explanation of its function and/or role in UPEC. Please describe this in short earlier in the text.

- An explanation of PapGII’s function was added in the abstract on L 22 and is explained in the introduction L 73.

Which samples are included in the 772 isolates that you mention in the abstract.

- The abstract has been rephrased to avoid confusion at L 21:

- “Here, we present a large-scale phylogenomic analysis of a spatiotemporally and clinically diverse set of 907 *E. coli* isolates, including 722 uropathogenic *E. coli* (UPEC) isolates.”

How was this selection made? Which collections?

- A paragraph on inclusion criteria for isolate collections was added to Supplementary Note 1.

Could you describe what type of *E. coli* was part of the fecal samples, did they share specific vir factors?

- Fecal isolates were not disease-associated and did not share specific virulence factors. Explained in Supplementary Note 1 (see also first comment).
- Virulence factors identified in the isolates are available in Supplementary Data 8.

How were the sublineages defined? Here, again hierBAPS can be a good tool.

- Sublineages were indeed defined using hierarchical clustering. We used fastBAPS (methods section L 581).

The samples that were confirmed papGII positive, did they harbour the whole operon? Where there any differences in the papGII operons in the papGII positive isolates?

- Forty-two *papGII* PAIs were fully resolved using long read sequencing and could be studied to address the variation of the *pap* operon. All contained complete *pap* operons. Sequence variation was however observed in the genes encoding P fimbrial subunits PapA and PapE, which are known to occur in varying alleles (results at L 159). A figure showing the sequence variation of the resolved *pap* operons was added (Supplementary Fig. 4).

What is needed for the P fimbriae to be expressed? Have the strains been phenotypically assessed for P pili expression?

- P fimbriae are encoded by eleven genes of the *pap* operon (L 158). Functionality of the P fimbriae can be assumed if the operon is intact, although the expression of P fimbriae is known to be subject to phase variation, regulated by a reversible epigenetic switch and by an sRNA-mediated mechanism (L 443).
- We analyzed genomic associations and did not perform phenotypic assays. This limitation of the study is acknowledged in the discussion (L 441).
- Functional expression of the P fimbriae during human urinary tract colonization has been shown recently (L 75 and L 356).

Did you identify the whole *afaVIII* operon? Did they differ from each other? Functionality?

- For isolates with resolved genetic context of *afaVIII* (17 of 18 *afaVIII*+ isolates), all contained whole *afaVIII* operons. *afaVIII* genes were highly conserved (>98% amino acid sequence identity), with exception of the gene coding for tip adhesin *afaEVIII* (>90% amino acid sequence identity). This is included to the results at L 201 and shown in Supplementary Fig. 10. We assume functionality as the operon is mostly complete and conserved.

If it is the whole operon that has been identified. I suggest you change it to *afaVIII* operon. You could also say that the isolates were positive for the *papII* operon and referred to as *papGII* positive in the text, and do the same for *afaVIII* positive isolates - refer them to as *afaEVIII* positive

- *afaEVIII* was changed to *afaVIII* throughout the manuscript.
- *pap* operons were named consistently.

Please describe what type of VAG this is for clarity. I can see you have done that in the section below but worth including its function when it's first mentioned.

- *agn43* encodes antigen 43, which is involved in biofilm formation, adhesion, and autoaggregation. This was added to line 206.

I am not completely sure I understand this section. Do they invasive UPEC have a higher prevalence of iron, immune, secretion systems, adhesion/invasion, toxins and bacteriocins related VAGS compared to ABU? Whereas invasive UPEC compared to cystitis isolates only had a higher frequency of VAGs related to iron and immune systems. Do I understand it correctly? Maybe simplify the section. This section is one whole sentence, split it into at least 2-3 sentences.

- This sentence was rephrased to clarify (L 217).

Phylogroup E/cladeI was not dominated by fecal isolates right? Supp table 2 only 1.1 % of the fecal isolates are in E/cladeI

- This was corrected (L 234)

Clarify that this is an operon of x number of genes.

- The *E. coli iuc* locus consists of six genes (added, L 356).

was the *iuc* locus complete in all *iuc* positive isolates?

- The *iuc* locus was complete in 92% of *iuc*+ isolates; This was explained in L 261.

“differed for their nature” What does this mean?

- This referred to their genetic composition and was rephrased, L 299.

Why do you think your dataset contains such a low number of CC131 UPECS?

- Our dataset contains many isolates collected before year 2000; the lineage CC131 emerged only recently (added at line 320).

fecal isolates: I am missing an explanation of why they were included and on what grounds they were included.

- See first comment and Supplementary Note 1.

I am not sure what this means. Do you mean that they carry certain virulence factors that may be specific for ExPEC?

- ‘ExPEC status’ was defined in the cited publication based on the presence of specific VAGs. This was added at L 521.

how were they cultured? What was used for DNA extraction? Culture of one colony or several?

- Overnight cultures grown in Mueller Hinton media obtained from single colonies were used for DNA extraction. Added to methods section (L 538)

Did you perform some sort of quality control of the samples? Like running Kraken/Bracken, Quast, CheckM.

- Quality metrics are provided in Supplementary Data 1. Quast was used for quality control (L 564).

How was the core genome identified?

- Core genome alignment based on collinear blocks were constructed using parsnp (further outlined, L 567).

Figure: Mentioned this before, could you add the ST in addition to the CC?

- Sequenced types were added to Figure 1 and referred to in main text.

Remind the reader what L1, L2 and L3 refers to.

- CC73-L1, L2, L3, and L4 refer to *papGII+* sublineages within CC73 (added to legend of Figure 5)

Some of the isolates are specifically indicated, clarify why and what these represent, like US03

- Isolates with high-quality assemblies used to investigate the genetic context of *papGII+* are indicated. This was added to legend of Figure 5.

I had to look at the figure a few minutes to clearly see which dots belonged to which ring. Maybe you could make it even more clear. Add a dotted line in between the different rings around the tree.

- Figure 5 was modified to make the differentiation between rings easier.

Suppl page 6: Make it easier for the reader and remind them of which strain this was.

- UMN026 is a reference strain according to which genes in the GWAS plot in Figure 2 are ordered. This is added to the legend of Supplementary Figure 21.

Suppl page 7: The studies with fecal samples, why were they isolated in those two studies? What was the question to be answered in the MN and KTE studies/collections?

- See first comment and Supplementary Note 1. All fecal isolates were originally used as control strains for genetic comparisons with UPEC isolates.

Suppl page 9: Based on *intB* and *intS*? Clarify which genes it was based on.

- The tree shown in Supplementary Figure 6 was based on *intB* and *intS* (added to legend).

Suppl page 10: I suggest adding the ST profile of the isolates as well.

- ST profiles were added to Supplementary Figure 16.

Suppl page 11: Are these loci complete, do you think that they are both functional?

- Both *pap* operons are complete and presumably functional (added to legend of Supplementary Figure 17).

Suppl page 12: Change to *afaVIII*. In the figure *afaVIII* is used to show the location.

- The annotation in the Supplementary Figure legend 11 was changed to *afaVIII*.

Suppl page 13: No info on the populations? Missing data? If so, could you add “not known” or something similar.

- The population investigated in the original study (general population, i.e. both male and female patients) was added to Supplementary Table 1.

Suppl page 24: Was there any difference between the papGII loci between the isolates representing the different clinical phenotypes, i.e. papGII from fecal samples vs papGII from Invasive UPEC isolates?

- Due to fragmented *pap* operons assemblies from short read data, a systematic analysis was not possible. Due to the limited variation observed in the *papGII* operon and the fact that UPEC have the gut as reservoir, we expect similar genetic compositions.

Reviewer #2 (Remarks to the Author):

Major Comments

As the manuscript currently reads – it is unclear at the end of the introduction why the *papGII* is the focus of this research.

- Our work uses a genome-wide approach to compare invasive versus non-invasive UPEC isolates, in which *papGII* was identified as key virulence gene. We added a clearer explanation of the aim of our study at the end of the introduction (L 80).

There is some text in the discussion which would be better in the introduction (lines 293-299)

- As the association of invasive UPEC with *papGII* was found as a result and we intended to place this finding in the context of previous studies, we believe it would remain best in the discussion part.

Further, the concept of pathogenicity islands and the role they play in virulence in bacteria is also not clearly addressed. This directly relates to the acquisition of putative virulence factors associated with UPEC and is important to background.

- An explanation of the concept of term pathogenicity islands, including their putative role virulence, was added to L 76.

Clinical terms such as pyelonephritis and cystitis need to be clearly defined as what these terms refer to. At present this reads as assumed knowledge which it can't be.

- An explanation of the terms cystitis and pyelonephritis was added to the abstract (L 24), introduction (L 33), and Supplementary Note 1.

The pathotype terms UPEC and ExPEC are used interchangeably through the introduction. This is incorrect. While all UPEC are ExPEC, not all ExPEC are UPEC. Please ensure these terms are defined and used appropriately.

- Definitions were rephrased to avoid confusion (L 38) and the introduction was carefully checked for appropriate use of the terms.

No mention is made of antimicrobial resistance in the introduction, which given AMR is addressed in the results and discussion, this needs to be briefly addressed.

- A brief background on AMR in *E. coli* was added to introduction L 62.

At the end of the introduction I was unclear on the specific gap in knowledge the authors were addressing and what the key research questions were for this study.

- We added a more detailed explanation of the aim of our study at the end of the introduction (L 80).

2. The study design was not clearly explained (both in the methods and the results). It was unclear why the 16 genome datasets (4 novel ones in this study and 12 others) were appropriate to address the research question.

- We added the rationale for the inclusion of isolate collections in Supplementary Note 1.

- Only limited genomic data of *E. coli* annotated with associated relevant phenotypes (ABU, cystitis, pyelonephritis, urinary-source bacteremia) are publicly available. All such identified public genomic collections were included in the study. The public data was further supplemented with genomes collected and/or sequenced as part of this study. The final dataset consisted of a spatiotemporally diverse samples with similar numbers of invasive and non-invasive UPEC isolates.

Are there sampling biases that need to be addressed? E.g. one of the new datasets MVA_{ST}_ABU was a subset of a larger study – how were these isolates refined? In the public data - line 427 states 35 isolates were excluded for sampling bias?

- We acknowledge that sampling bias is intrinsic to the availability of public data (L 437).
- The original MVA_{ST} collection included *E. coli* isolates from various clinical sources (e.g. urine, blood, respiratory). For this study, we selected urinary isolates that could confidently be assigned to asymptomatic bacteriuria. L 497 was rephrased to clarify.
- 35 isolates from the PUTI_cys, MC_pye, and MN_fec collections were excluded to correct for sampling bias introduced during the original selection for sequencing. Aim of the source study was that approximately half of all sequenced isolates are ExPEC-marker positive. For our study, 35 isolates were randomly excluded to restore the originally reported ExPEC-marker presence:absence ratio of the respective source collection (L 519). The method section L 507 to L 526 was rephrased to clarify.

Most of the data in this study is public data. It is great to see public data used in this way, however further detail of why these data were chosen and not others is required. What were the selection parameters for including these public data and how might it impact upon the study design?

- Information on why these collections were included is added in the Supplementary Note 1.
- Criteria for the inclusion of UPEC isolate collections were the availability of associated metadata on clinical syndromes and/or medical diagnosis, i.e., asymptomatic bacteriuria (ABU), cystitis, pyelonephritis, urinary-source bacteremia, or urosepsis. See also previous comment.

The inclusion of a table in the main body of the manuscript may be useful to the reader to understand where and when the data has been collected. This could include study, number of isolates, country of isolation, timespan of samples, clinical presentation associated with the samples, age of the patients (looks as though most were >60 years), and refs to the studies for the public data. There are some figures in the supplementary that try to address but I think consolidating into one table may help

- Table 1 was added including a brief overview of the included isolate collections. Further information on the collections is included as Supplementary Table 1.

3. Inclusion of the ~180 faecal isolates. I am unclear on what these isolates bring to the study and also the sampling strategy for their inclusion. They are already excluded from the GWAS analyses in this study. At times they are used as comparative reference for rates of genes present/absence in invasive v

non-invasive UPEC. However, it doesn't appear these faecal associated isolates were selected to be broadly representative of the diversity of *E. coli*. These data could be removed from the study which would help to simplify the study or their inclusion needs to be appropriately addressed.

- See first comment reviewer 1 and Supplementary Note 1. These isolates were included as context for the genomics analyses.

4. Long read assemblies – further detail is required

How/why were isolates selected for long read sequencing?

- De novo assembly from short read Illumina sequencing data did not allow reconstruction of complete genomes and specific regions such as the *papGII* PAIs might be incomplete. Therefore long-read sequencing was done on representative isolates from dominant *papGII*+ lineages. We aimed to have minimal one isolate of each *papGII*+ lineage with a high-quality genome assembly available. This was clarified in L 153.

Were default settings used in all cases for both HGAP and Unicycler?

- Default settings were used (added to methods section L 555).

Was there any selection for long reads based on length?

- Selection of reads is implemented in the HGAP and Unicycler pipelines as default.

The authors note that plasmids were lost from HGAP approaches that were obtained from Unicycler. Please provide details of the long read assemblies so the reader can tell what assembler was used to generate the data. This could be provided in a supp table with info for each chromosome and plasmid in each isolate in addition to accession numbers for the complete genomes generated in this study.

- Assembly method and accession number of each isolate and contig are provided in Supplementary Data 5 and 6.

5. Screening for genes in assemblies and short read data. Please state if default parameters were used or if changed to ensure reproducibility.

- Parameters (default) were added to the methods section.

6. Phylogenetic analysis. Please give extra details about what references were used. Were different alignments made for each CC specific analysis? Line 479 states reference genomes had phage regions masked. Reference name and accession would address this.

- Different alignments were made per CC specific analysis (L 569 and legend of Fig. 5). Reference names and accession IDs were added. Phage regions as identified with phaster in the respective reference genomes were removed (L 573).

7. I have some concerns about how CC73 was split into four sub-lineages by using a patristic distance in RAMI that was different to that used for everything else. What were the grounds for this lineage being further split?

- The population structure and *papGII* acquisitions in CC73 show a higher diversity than other CCs. To address this, sublineages were investigated on an additional level (L 548). The suffixes 'L1' to 'L4' were used for CC73 *papGII*+ sublineages to indicate this second level of

hierarchy. This was also added to the legend of Fig. 3 and explained in the results section L 148.

8. Manhattan plots – Fig 2 and Supp Fig 4. These were confusing to interpret initially as they have the same figure title and same figure legend until the second half there the difference in what is being shown is stated. For Fig 2 – this needs to be clear that only showing data for 4764 COGs that are part of the reference genome in the title. The same applies for Supp Fig 4 but in reverse which shows the remaining 25941 COGs.

- The title of Supplementary Figure 21 and Figure legends were modified.

9. Characterisation of PAIs. A total of six PAIs with *papGII* are characterised in this study from the 35 complete genomes. More analyses in characterising the PAIs would enhance one of the main findings of this study.

- We added Supplementary Figure 22, showing the genetic composition of all 42 resolved PAIs from the 35 complete genomes. Annotated sequences of the PAIs are available on https://github.com/MBiggel/UPEC_study.
- The 42 PAIs could be clustered into six types based on sequence similarity. Representative PAIs of each type are shown in the main body (Fig. 3).

Fig 3 – in the 14 lineages with *papGII* – not all lineages are shown as investigated for PAI. Why not?

- In Fig. 3, only PAI characteristics of dominant *papGII* lineages, i.e., with >5 isolates, are shown (figure legend). Collapsed branches of dominant lineages were colored to clarify.
- PAI characteristics per *papGII+* isolate, including isolates not falling into these dominant lineages, is provided in Supplementary Figure 16.

From the figure it appears that ~50% of PAI in the 14 main lineages were characterised. This significantly limits what you can infer about PAI acquisition and maintenance in the different lineages. Fig 3 - For the *papGII* lineages where no PAI were reported and those such as CC73 where large number of isolates are shown in grey meaning not determined – were these investigated further?

- PAI identification was based on either long-read sequencing or similarity with reference PAIs using read-mapping based typing, and not possible to confidently assign for all isolates with our data (shown in grey). This is described in the methods section L 649 and was clarified in legends of Figures 3 and 5.
- *papGII+* PAI characteristics of single isolates and small lineages (1-4 isolates) were not reported in Figure 3 but are shown in Supplementary Figure 16 and Supplementary Data 7.

Is it possible to try and characterise these novel lineages from the assembly data – at least of the immediate *papGII* region? E.g. in CC14 the tRNA site was identified but was a novel variant. Is it possible to characterise these PAIs?

- We characterized all *papGII+* PAI types and insertion sites that could be resolved. However, this was not possible for all isolates due to fragmented *papGII+* PAIs from short-read data (see previous comment).

- Genetic characterization of a CC14 PAI and its novel insertion site is provided in Fig. 3 (PAI_{US12-gnl}), Supplementary Figure 22, and Supplementary Figure 5b (integration site *gln*).

The nomenclature of having the tRNA site included in the name of the PAI with papGII is difficult – especially where integration was at a different tRNA site. E.g CFT073-pheU but in fig 3 the most common site was pheV. Suggest using -1, -2 etc to delineate different PAI with papGII from same genome.

- To improve readability, PAI clusters were renamed to type I to VI throughout the manuscript.

10. Fig 4 – PAIs and VFs.

Difficult to see the detail on panels B and D. Most of the info in panel C is in Fig 3. Could consider a restructure to highlight key results.

- Key results from Figure 4 were combined with Figure 3 and remaining panels moved to the Supplementary Figures 8 and 9 with increased figure size.

How does the six families of PAIs related to the core genome structure?

- A tanglegram comparing the core genome phylogeny to the sequence similarity of PAIs was added (Supplementary Figure 7).
- The distribution of the six PAI families across the phylogeny (i.e., core genome similarity) is provided in Figure 9 and, in more detail, Supplementary Figure 16.
- In addition, the tree in Supplementary Figure 8 was annotated with clonal complexes to relate to the core genome structure.

11. Were different CCs characterised by different complements of VFs? How might this impact clinical outcome (particularly as not all papGII positive isolates were invasive UPEC)? The *iuc* gene is explored in a little detail but only in three CCs. Also – line 185 states ‘significantly more VAGS’ – this implies statistically significant – correct?

- We added Supplementary Figure 14 showing the prevalence of main VFs by CC. Within a given CC, VFs showed a similar distribution among invasive and non-invasive UPEC (L 250).
- The distribution of *iuc* in the phylogeny including all CCs is shown in Figure 4b. *iuc* alleles and associated mobile genetic elements are described in detail in Supplementary Figure 15 and L 263.
- Overall, VAGs were significantly more abundant among invasive UPEC isolates than in non-invasive UPEC isolates. However, after accounting for population structure, there was no statistically significant difference in the number of VAGs between invasive UPEC and cystitis isolates. These findings are part of the manuscript at L 217 and L 236.

12. Three pandemic ExPEC lineages.

Why the switch in the language to ExPEC when been using UPEC?

- ‘Pandemic ExPEC lineage’ was changed to ‘pandemic UPEC lineage’ (L 272).

Colours in Fig 6 – please use different colours – e.g. grey is used for both *iuc* presence and faecal isolates. PAI and cystitis are very similar shades of yellow. Also what is the significance of pink shading (not the red for *papGII*)?

- Figure 5 was modified and the shading was explained in more detail.

13. Role of *afa* - Line 169 – states *afa* has a similar role in (putative) pathogenesis of *papGII*. What is the basis for this claim? Could be a hypothesis (for discussion) but the results here don't show this. This is repeated in the discussion at lines 299-301.

- The gene *afaVIII*, associated with pyelonephritis in previous studies, was here identified in a lineage that was significantly associated with invasive UPEC isolates but lacked *papGII* (L 192). Its potential role in invasive UTI is a hypothesis that requires further investigation. This hypothesis was removed from the results part (L 201) and phrased more carefully in the discussion part (L 361).

CC95 – lines 233-234 – states there has been excision and reintegration of the PAI. What was the supporting evidence for this? Supp Fig didn't address how could make this statement for the CC.

- Supplementary Figure 16 shows that among closely related CC95 isolates, the same PAI (Type V) was identified at either the *pheV* or the *pheU* site. We can indeed not state the actual excision and integration, and L 282 was rephrased as a suggestion.

CC73 – *iuc* gene – alternative hypothesis is that the *iuc* gene was acquired and then largely maintained within the CC73 lineage with different PAIs then acquired.

- The two distinct *iuc*-containing PAIs identified in CC73 might originate from two independent acquisition events or have evolved within CC73 from a common ancestor after acquisition of a single PAI. The hypotheses were both raised at L 315.

Unclear how different the sub-lineages of CC73 were – both CC73-L1 and CC73-L4 have <50% of the isolates characterised for PAIs.

- See earlier comment. A random selection of isolates was not characterized for their PAIs due to fragmented genome assemblies. Isolates of CC73-L1 and CC-L4 with identified *papGII*+ PAI types showed different types. This and the population structure support to differentiate them in separate sublineages.

Further - was any investigation undertaken of the genomic context of *iuc*? Was this the same in CC73 from different sub-lineages or different? Potentially acquired once and then lost in a handful of isolates?

- The genomic context of *iuc* alleles was resolved and associated mobile genetic elements are shown in Supplementary Figure 15. *iuc* was part of plasmids or specific pathogenicity islands (L 263).
- A more detailed description of *iuc* mobile genetic elements in CC73 was added to L 310. Three of the four *papGII*+ sublineages (-L1, L2, L4) shared the same *iuc*+ PAI; the fourth

sublineage (-L3) carried a *iuc+* PAI with a different *iuc* allele and lacking *papGII* (Fig. 3 and L 312).

papGII in non-invasive isolates – lines 331 – due to host being ‘relatively resistant’ to invasive UTIs. What is the data to support this statement and what does ‘relatively resistant’ mean?

- Previous studies, summarized in a review cited at L 394, showed varying host resistance to UTI. ‘Relatively resistant’ was rephrased to ‘more resistant (L 395).

Further, no data are presented on the *papGII* interacting with host cells. While data presented suggests *papGII* may be associated with the invasive phenotype, and a previously study has shown some kind of functional role of *papGII*, it is worth noting that this study only presents genomic data. The functionality of *papGII* in the different CC backgrounds has not been shown, much less that this is the definitive pathogenesis mechanism e.g. lines 120-121 - ‘specifically genetically determined pathogenesis mechanism’ for invasive UPEC.

- This is clearly mentioned and acknowledged as a limitation of this study (L 441).
- The sentence ‘specifically genetically determined pathogenesis mechanism’ was removed from the results section (L 141).

14. Structure of results. The results section felt disjointed and it was difficult to follow at times.

Suggest a restructure of the results so clear from each section what the main findings were and this would be improved by having full paragraphs rather than a series of very short blocks of text. This would greatly improve the readability of the manuscript. e.g. results on distribution of *papGII* is split over 9 paragraphs – the shortest on being 2 sentences and 4 lines (lines 171-174).

- The results section has been edited to improve structure. Short paragraphs have been combined to improve readability, including lines 171 – 174 in the original submission.

15. Future work – this study presented some interesting data which could form the basis for future genomic and wet lab based experiments. Any considerations of what would be important research questions rising from these data?

- Potential future research work has been added (L 380).

16. Use of supplementary tables / figures. Please ensure there are referred to in order in the manuscript. For the tables went from supp table 1 to supp tables 9 and 10 and with supp figures from 1 and 2 to supp fig 5.

- The order of Supplementary Figures has been corrected.

Minor Comments

Please check language for terms such as ‘non-demented male patients’ (line 419). I was unsure what this meant. If this is a medical term it needs a clear definition.

- This term has been modified (L 495) to male patients.

Great to see the data on microreact so ppl can explore it interactively. A minor point is that some of the countries from Korea, France, Germany, Australia and Mexico aren't shown on the map as the latitude and longitude data are included in table to better showcase the diversity of countries included in this study

- Data was added to microreact. Also reference strains with available geographic origin are now labelled.

Fig 4 – please label X axis for barplots – with proportion 0, 25, 50 75 and 100.

- A barplot X axis scale was added to Figure 3.

Reviewer #3 (Remarks to the Author):

Specific comments:

1. The clinical diagnosis of pyelonephritis is highly subjective and its use in the definition of “invasive” disease may be unreliable.

- Pyelonephritis isolates originate from a study by Talan et al. (Table 1). The diagnosis of uncomplicated pyelonephritis was based on flank pain/tenderness, fever (>38.0 °C), and pyuria, in line with EAU guidelines on Urological Infections (Supplementary Note 1). In addition, only premenopausal women without immunocompromised conditions, diabetes, or urological abnormalities were included to avoid misclassifications.

2. The authors acknowledge that the absence of well-documented information about host-related predisposing medical conditions may have caused misclassification of invasive UPEC strains. In the Methods, the authors state that for the urosepsis isolates, they excluded isolates from patients who were immunocompromised, pregnant, had surgery or urological intervention, and had putative hospital-acquired infection. However, they also state that the median age of the patients from which the urosepsis isolates were analyzed was 75.5 years. Most cystitis cases would be expected to occur in a much younger age group. Age itself, regardless of predisposing conditions, is a major determinant of invasive disease. What was the median age of the patients who had non-invasive UTI? Without controlling for age, it is misleading to attribute *papGII* to so-called “invasive UPEC”.

- To address this concern, we analyzed the prevalence of *papGII* among invasive vs non-invasive UPEC stratified by age groups (Supplementary Table 4a). The significant association of *papGII* with invasive UPEC was observed across all age groups (L 132).
- The median host age of the collections was added to Table 1 (when metadata available).

3. Same for gender. Cystitis is much more frequent in women while invasive disease is similarly distributed in both genders, especially older patients. Was prostatitis included in the definition of invasive UTI?

- We analyzed the prevalence of *papGII* among invasive vs non-invasive UPEC stratified by host gender (Supplementary Table 4b). The significant association of *papGII* with invasive UPEC was observed in male and female hosts (L 132). We did not have access to prostatitis isolates to investigate to prevalence of *papGII*.

4. The study does not describe the potential association of any of the antimicrobial drug-resistance (AMR) genes or a set of such genes with the two clinical phenotypes. Infections caused by UPEC strains containing AMR genes conferring resistance to broad-spectrum antibiotics (e.g., ESBL-producers) or strains containing multiple AMR genes lead to greater morbidity and complications due to incorrect drugs or delayed initiation of a correct treatment regimen.

- We acknowledge that drug-resistance may impact the severity of infections. However, resistant as well as largely susceptible pandemic lineages are seen as an important cause of invasive UTI. A remark was added to the discussion at L 411.

- With GWAS, we used an unbiased approach to test all variants and genes of the pan-genome for significant associations, including AMR genes (L 123 and L 126). A potential underestimation of rare variants or of the combined effect of variants is discussed in L 428.

5. In the Discussion, the authors speculate that, in addition to its role in invasive pathogenicity, this *papGII* may be involved in transmission via clonal expansion in animal niches other than the human host, such as poultry. Since ExPECs are frequently isolated from poultry, it would be interesting to know if *papGII* is overly represented in such isolates.

- Different studies showed that *papGII* is overrepresented among avian pathogenic *E. coli* from poultry (*papGII* prevalence 40 – 60%). This was added to the discussion (L 428).

6. Table 1 shows % frequency of *papGII* in invasive vs non-invasive UPEC isolates. It would be more informative to show numbers also (number *papGII*/total).

- Numbers were added to Table 2.

Reviewers' Comments:

Reviewer #1:

Remarks to the Author:

Dear authors,

I have read the updated manuscript. You have taken all comments and suggestions I had in to consideration. I am happy with your response to those as well as to you changes to the text as well as the figures.

I only have a few additional comments which you can see in the attached pdf.

Reviewer #2:

Remarks to the Author:

I have reviewed the revised version of this paper and the responses to the initial critique and comments. I appreciate the additional information and analyses undertaken by the authors and would like to thank the authors for their responses which have largely addressed my criticisms.

Minor comments

Introducing PAIs in the introduction. This is very limited and while I understand wanting to expand upon this is the discussion – the authors haven't stated clearly that papGII are carried on PAIs. Given this is a key result – it is important to make this clear to those that may not be familiar with PAIs in *E. coli*

The importance of AMR was limited in both the introduction and the discussion. The authors have only cited three papers and there are several key studies that could be cited and the discussion expanded a little more.

Supp Fig 18 – ExPEC in the title

Reviewer #3:

Remarks to the Author:

Revision:

This revised paper addresses the major concerns raised by this reviewer. The Discussion now describes limitations of the observations. Most "invasive" disease following cystitis results from prolonged infection, and the suggestion by the authors that papGII may confer niche-specific adaptation would be consistent with such infection outcomes. However, since papGII is also frequently found in avian *E. coli*, a possibility has to be entertained that the human gut gets colonized by *E. coli* in food (e.g., poultry products) and that these invasive *E. coli* strains containing papGII are not necessarily human gut commensal *E. coli*, as the authors seem to suggest.

REVIEWERS' COMMENTS

We thank the reviewers for their in-depth reading and constructive suggestions to improve our manuscript. The adaptations made to the manuscript are listed below, referring to the line numbers (L) in document “Manuscript revision 2 changes tracked”.

Reviewer #1 (Remarks to the Author), comments in the pdf file:

L 41: A very long sentence.

- This sentence is rephrased and split in two (L 45).

L 56: Not only commensals belong to A and B1, other *E. coli* pathotypes as well.

- This sentence was rephrased to include referring to intestinal pathogenic *E. coli* (L 62).

L 85: What context, need to elaborate more here to describe the specific reasons for including these and specifically these fecal samples.

- The type of *E. coli* (non-disease associated) and reason for inclusion were included at L 87.

L 116: associated with invasiveness instead?

- The sentence was rephrased to “associated with invasiveness” (L 116).

L 174: change to: papGII gene clusters or papGII operons (if it has been defined as an operon previously)

- This was rephrased to *papGII* operon (L 168) and the operon was defined before (L 152).

L 178: Should this be papGII operon or do you refer to other types of pap subtypes that have been identified?

- This was changed to *papGII* operon (L 170).

L 185: maybe consider using types here? Or maybe, PAIs could be subtyped into six types, I through VI.

- “Category” was changed to “types” (L 176).

L 192: This section can shorten. First and last sentence says the same thing.

- This section was shortened (L 183).

L 277: not needed.

- The percentage was removed (L 260).

L 345: as well as from feces

- This suggestion was added (L 321).

L 353: The adhesin papGII has been...

- This suggestion was added (L 329).

L 361: change to: identify a comparative loci, specifically *afaVIII*, as a potential marker of invasive UPEC.

- This was rephrased to “our data also identify a comparatively rare locus, specifically *afaVIII*, as potential marker of invasive UPEC” (L 337).

L 646: Reference for BLAST+

- A reference for BLAST+ was added (L 622).

Reviewer #2 (Remarks to the Author):

Minor comments

Introducing PAIs in the introduction. This is very limited and while I understand wanting to expand upon this is the discussion – the authors haven’t stated clearly that papGII are carried on PAIs. Given this is a key result – it is important to make this clear to those that may not be familiar with PAIs in *E. coli*

- We added a sentence to the introduction stating that *papGII* is part of PAIs (L 79).

The importance of AMR was limited in both the introduction and the discussion. The authors have only cited three papers and there are several key studies that could be cited and the discussion expanded a little more.

- We expanded on the discussion on AMR in UPEC and cited additional papers (L 387).

Supp Fig 18 – ExPEC in the title

- In the legend of Supplementary Fig. 18, “ExPEC lineage” was changed to “UPEC lineage”.

Reviewer #3 (Remarks to the Author):

However, since papGII is also frequently found in avian *E. coli*, a possibility has to be entertained that the human gut gets colonized by *E. coli* in food (e.g., poultry products) and that these invasive *E. coli* strains containing papGII are not necessarily human gut commensal *E. coli*, as the authors seem to suggest.

- *papGII+* isolates are indeed not necessarily commensals and human intestinal acquisition of *papGII+* *E. coli* isolates might occur through contact with animals or animal products, in addition to direct transmission between humans. We added a statement on potential transmission routes of ExPEC/UPEC (L 403).
- PapGII might indeed contribute to the colonization of an animal reservoir rather than the human gut (L 401).